# Security-Informed Safety Analysis of Autonomous Transport Systems Considering AI-Powered Cyberattacks and Protection

**DOI:** 10.3390/e25081123

**Published:** 2023-07-26

**Authors:** Oleg Illiashenko, Vyacheslav Kharchenko, Ievgen Babeshko, Herman Fesenko, Felicita Di Giandomenico

**Affiliations:** 1Department of Computer Systems, Networks and Cybersecurity, National Aerospace University “KhAI”, 17, Chkalov Str., 61070 Kharkiv, Ukrainev.kharchenko@csn.khai.edu (V.K.); h.fesenko@csn.khai.edu (H.F.); 2Software Engineering & Dependable Computing Lab, Istituto di Scienza e Tecnologie dell’Informazione “Alessandro Faedo”, Area della Ricerca CNR di Pisa, Via G. Moruzzi 1, 56124 Pisa, Italy; felicita.digiandomenico@isti.cnr.it

**Keywords:** autonomous transport system, unmanned aerial vehicle, unmanned maritime vehicles, artificial intelligence, AI-powered attack, IMECA, SISMECA

## Abstract

The entropy-oriented approach called security- or cybersecurity-informed safety (SIS or CSIS, respectively) is discussed and developed in order to analyse and evaluate the safety and dependability of autonomous transport systems (ATSs) such as unmanned aerial vehicles (UAVs), unmanned maritime vehicles (UMVs), and satellites. This approach allows for extending and integrating the known techniques FMECA (Failure Modes, Effects, and Criticality Analysis) and IMECA (Intrusion MECA), as well as developing the new SISMECA (SIS-based Intrusion Modes, Effects, and Criticality Analysis) technique. The ontology model and templates for SISMECA implementation are suggested. The methodology of safety assessment is based on (i) the application and enhancement of SISMECA considering the particularities of various ATSs and roles of actors (regulators, developers, operators, customers); (ii) the development of a set of scenarios describing the operation of ATS in conditions of cyberattacks and physical influences; (iii) AI contribution to system protection for the analysed domains; (iv) scenario-based development and analysis of user stories related to different cyber-attacks, as well as ways to protect ATSs from them via AI means/platforms; (v) profiling of AI platform requirements by use of characteristics based on AI quality model, risk-based assessment of cyberattack criticality, and efficiency of countermeasures which actors can implement. Examples of the application of SISMECA assessment are presented and discussed.

## 1. Introduction

### 1.1. Motivation

Cyberattacks have emerged as significant threats that can severely impact the safety of critical infrastructures, industrial control systems, and transport [1,2,3]. Attacks refer to exploiting vulnerabilities within systems, software, or networks with malicious intent. From individual hackers to sophisticated hacker centres, adversaries constantly seek to identify and exploit vulnerabilities to gain unauthorised access, steal sensitive information, disrupt operations, or even cause physical harm. The consequences of successful attacks can be devastating, leading to financial losses; reputational damage; and, most importantly, safety compromise.

This issue is challenging for autonomous transport systems (ATS), such as unmanned aerial vehicles (UAVs), driverless cars, unmanned maritime vehicles (UMVs), satellites, and so on. The application of artificial intelligence (AI) as a means for improving the efficiency of cyberattacks [4] and protection of ATS assets is an additional extremely important direction of research and development in the context of security and safety. Studying and analysing different attack vectors, techniques, and associated vulnerabilities is needed to understand the risks various ATSs face and develop effective countermeasures considering AI-powered attacks and protection issues.

This paper aims to contribute to the growing body of knowledge in the field by examining the specific effects of cybersecurity attacks on safety, focusing in particular on ATSs in different domains, enhancing the Security-Informed Safety (SIS) approach and introducing the SISMECA (SIS-based Intrusion Modes, Effects and Criticality Analysis) technique aimed at systematising threats, vulnerabilities, attacks, and safety risks, as well as at reducing criticality by providing countermeasures for identified critical failures. This technique is added considering the analysis of artificial intelligence influence on ATS security and safety using decomposition of AI quality characteristics set to minimise uncertainties/entropy of assessment. Three kinds of ATSs are detailed in the investigation of safety and security: UAVs, UMVs, and satellites.

### 1.2. State of the Art

Keeping in mind the SIS approach mentioned above, the analysed references were subdivided into the following:the references describing stages of the Modified Cybersecurity Kill Chain for AI-powered attacks [5,6];the references addressing AI-powered attacks utilised against various components of the ATS infrastructure [5,6,7,8,9,10,11,12,13];the references presenting ATSs cybersecurity assessment and assurance techniques, including decisions based on the application of AI and AI platform (AIP) [14,15,16,17,18,19,20,21,22,23,24,25,26,27,28,29,30,31,32,33,34];the references highlighting ATS safety issues in the context of the ATS cybersecurity assessment [35,36,37,38].

The groups of publications mentioned in this list are further detailed in reference analysis Table 1 and Table 2; then, explanations for each of these references is also provided.

Kaloudi et al. [5] described the Modified Cybersecurity Kill Chain as having five stages and allocated classes of AI-powered attacks among them in the following ways, based on the study [4]:Access and penetration (Acc&Pen) stage. This stage covers automated payload generation/phishing, password guessing/cracking, intelligent capture attack/manipulation, smart abnormal behaviour generation, and AI model manipulation.Reconnaissance (Rec) stage. This stage involves intelligence target profiling, smart vulnerability detection/intelligent malware, intelligent collection/automated learn behaviour, and intelligent vulnerability/outcome prediction.The delivery (Del) stage. This stage entails intelligent concealment and evasive malware.The exploitation (Exp) stage. This stage addresses intelligent lateral movement and behavioural analysis to identify new exploitable vulnerabilities.The command and control (C&C). This stage deals with intelligent self-learning malware, automated domain generation, and denial of service attacks.

AI-powered attacks which can be utilised against the ATS infrastructure are presented in Table 1. In particular, on-board equipment (AI-based/non-AI-based equipment), ground control station (GCS) equipment (AI-based/non-AI-based equipment), and two channels (ATS-ATS and ATS-GCS) are considered. For each attack, both the technique used for its implementation and the stage at which it is applied are indicated. The techniques used are as follows: Neural Networks (NNs), Generative Adversarial Network (GAN), Recurrent Neural Network (RNN), Logistic Regression (LR), Support Vector Machine (SVM), Support Vector Classification (SVC), Random Forest (RF), K-Nearest Neighbour (KNN), Gradient Boosted Regression Trees (GBRT), K-means clustering, and Deep Neural Network (DNN).

Table 1 reveals that (1) a non-AI-based GCS is the most vulnerable component of the ATS infrastructure to AI-powered attacks, (2) vulnerability prediction and self-learning malware are the most common types of attacks on components of the ATS infrastructure, (3) the most significant number of AI-powered attacks (free attacks) occurs at stage Acc&Pen, and (4) none of the analysed papers paid due attention to safety issues.

A considerable number of publications are connected to ATSs cybersecurity assessment and assurance techniques, including decisions based on the application of AI and AIP. They can be divided into the following groups: (a) application of AI for the cyber defence of ATSs and cases dealing with cyber incidents and potential threats to ATS assets [14,15,16,17,18,19,20,21,22,23,24] (Table 2), (b) methods and techniques of ATS cybersecurity analysis and assurance [25,26,27], and (c) AI quality models in the context of cybersecurity assessment assurance [28,29,30,31,32].

Table 2 reveals that (1) all significant types of ATSs (UAVs, satellites, MASSs) are vulnerable to cyberattacks, (2) UAVs and satellites are increasingly using AI-based intrusion detection/prevention solutions to withstand cyberattacks, (3) DL-based and CNN-based intrusion detection/prevention solutions are the most common for UAVs and satellites, and (4) the utilisation of AI-based intrusion detection/prevention solutions for MASS has not yet become widespread enough.

To assess the consequences of unauthorised intrusion into IoD systems, the Intrusion Modes, Effects, and Criticality Analysis (IMECA) method was utilised [25]. The authors of [26] presented a solution to automate the FMECA process for complex cyber-physical systems to reduce and mitigate the number of critical faults. Both a theoretical and mathematical model and a method to analyse the internal components of a security system and access to assets were suggested in [27]. The technique allowed for the analysis of asset security through the use of environment variables and physical security controls of the facility.

Principles of AI quality model development, as well as the procedure for realising its hierarchical construction considering trustworthiness, explainability, cybersecurity, and other characteristics, are described in [28,29]. The paper [30] suggested a quality model that allowed for objectively specifying and assessing qualities for ML systems based on an industrial use case. An example of a quality model building for a concrete industrial use case was given, and lessons learned from applying the construction process were analysed and discussed. A metric-based assessment technique of AI system (AIS) quality using radial diagrams is described in [31]. The study in [32] examined terminology and challenges in quality assurance for AI-based systems and characterised AI-based systems along the three dimensions of artefact type, process, and quality characteristics.

Thus, the cybersecurity domain for ATS encompasses the issues (both challenges and opportunities) considering the application of AI means, which should be analysed and assessed. The system has vulnerabilities that lead to the risk of being exploited, causing operational impacts. In this case, it is necessary to systemise and define requirements for AI means, assess the risks of critical failures, and mitigate risks by proper choice of countermeasures [33,34]. However, it is also important to note that ATS cybersecurity assessment should be performed considering safety issues [35,36,37,38]. The authors of [36] propose an STPA (System-Theoretic Process Analysis)-based methodology for MASS safety and security assessment, namely, STPA-SynSS. To continually track and manage hazards, the STPA-SynSS provides a comprehensive process for identifying risks, revealing causal factors, and implementing hazard elimination/mitigation strategies into the system design via system safety and security requirements. The work [37] addresses a methodological framework for the risk assessment of drone intrusions in airports, tailored to drone intrusion features, airport features, and current operations, and considering both safety-related and security-related causes. The study [38] identifies the vulnerabilities that AI systems may introduce to satellite networks and analyses the potential operational threats and effective technological and regulatory mitigation measures. Even though the analysed references emphasise the necessity of ensuring ATS safety with cybersecurity issues in mind, no real techniques for assessing such an impact are offered. That is, there is no systematic risk assessment approach at the IT (cybersecurity) and IoT (safety) levels.

Safety is of paramount importance in the context of the cybersecurity of ATSs due to several reasons:Human lives at stake: ATSs are designed to operate without direct human control, thus placing a significant responsibility on ensuring the safety of ordinary citizens and users of other ATSs/non-ATSs. Any compromise in cybersecurity could result in accidents, injuries, or even loss of life.Public trust and acceptance: The successful implementation of ATSs relies on public trust and acceptance. If safety is not prioritised and incidents occur frequently, it can erode public confidence in the technology. To gain widespread adoption, cybersecurity measures should be effective and transparent enough to demonstrate that ATSs are at least as safe, if not safer, than human-operated vehicles.Legal and regulatory compliance: Governments and regulatory bodies play a crucial role in establishing rules and regulations for ATSs. The cybersecurity measures applied should align with the safety standards and guidelines. Failing to meet these standards can result in legal and regulatory consequences, including restrictions or bans on deployment.Reputation and liability: Reputation management is essential to businesses creating and utilising ATSs. Safety incidents can severely damage a company’s reputation, resulting in financial losses and a loss of market share. Additionally, when accidents happen, liability issues arise, and figuring out responsibility can be problematic in the absence of explicit safety procedures.Ethical considerations: ATSs often face challenging ethical decisions, such as determining how to prioritise the safety of one group of people versus another in potential collision scenarios. Safety protocols need to address these ethical dilemmas in a transparent and accountable manner to ensure that the systems make the most ethically sound decisions possible.

Given these reasons, ensuring safety in the context of cybersecurity of ATSs is crucial for protecting human lives, gaining public trust, complying with legal requirements, preserving reputation, and addressing ethical considerations.

### 1.3. Objectives and Contribution

The main objectives of the investigation conducted in this work are as follows:To suggest a methodology, principles, and stages of Security-Informed Safety (SIS) and AI Quality Model (AIQM)-based assessment of autonomous transport systems, considering the application of artificial intelligence as a means to increase the power of cyberattacks and protect ATS assets;To develop modifications of known FMECA/IMECA (XMECA) techniques (table-based templates, algorithms of analysing failures/intrusions modes and effects criticality, criticality matrixes, and sets of countermeasures applied by various actors), their integrated option called SISMECA, and the corresponding ontology model;To investigate user stories related to ATSs in three domains (maritime, aviation, and space), connected with the application of AI for providing security and safety by use of AIQM and SISMECA-based techniques;To analyse the influence of countermeasures to decrease risks of successful attacks and ATS failures in the context of the SISMECA approach;To discuss the application of AIQM and SISMECA-based techniques to develop a roadmap of AI cybersecurity for ATSs in the context of the SIS approach.

The main contribution of the research is the development of a methodological base and techniques for providing an assessment of cybersecurity and safety that depends on cybersecurity and reliability of ATSs, considering the means of AI-powered attacks and protection. The suggested AIQM and SISMECA-based techniques allow for decreasing the uncertainty of risk assessment, due to the decomposition of safety and AI quality attributes, and minimising entropy measures. The methodology that is proposed develops and integrates several important things from a theoretical point of view. First, the principle of SIS was detailed and improved through the technique proposed by SISMECA, which ensures transparency of the assessment the consequences of cyberattacks. Secondly, combining this technique with an adapted AI quality model made it possible to implement SIS-based assessment of safety risks, taking into account the functions performed by AI tools (implementing the functions of ATSs or strengthening the protection of cyber assets).

This paper is organised as follows. Section 2 presents the methodology of research and the SIS approach to ensure the safety of ATSs, based on the AIQM and SISMECA techniques. Section 3 discusses examples of their application to analyse the cybersecurity and safety of ATSs. Section 4 describes the application of the mentioned techniques to develop a roadmap of AI cybersecurity for ATSs considering new challenges such as developing and using generative AI. Section 5 and Section 6 discuss the main results of the investigation and present conclusions and future research, respectively.

## 2. Methodology

As shown in Section 1, traditional cybersecurity and safety assessment methodologies must be revised due to the abovementioned challenges in today’s rapidly evolving ATS technological landscape. To address these challenges and therefore improve safety assessment, new methodologies should focus on the SIS approach and yet consider models of cutting-edge technologies such as AI.

This section provides a general scheme of the proposed methodology, highlights the SIS approach in the context of entropy, describes FMECA and IMECA as basic approaches used in SISMECA, and provides templates for all techniques and sample criticality matrices.

The place of SISMECA, among other XMECA-based techniques, is shown through an appropriate ontology.

### 2.1. General Scheme

The methodology of this investigation is based on the following consistently implemented principles (Figure 1):Specification of the SIS approach to the assessment of the ATS. The safety assessment is carried out considering the security threats and the analysis of the consequences of cyber intrusions for the functional safety of the ATS.Development of the modified and extended XMECA technique [39] considering the SIS/CSIS approach called SISMECA to assess the ATS security and safety step by step. The evolution from FMEA to SISMECA is illustrated by suggested ontology and templates. The SISMECA technique is a natural modernisation and generalisation of XMECA.User story/scenario-based SISMECA risk-oriented analysis of ATSs. Such analysis allows step-by-step formalisation of verbal information about incidents when AI has been applied or could be applied to protect cyber and physical assets.Decreasing risks considering:utilising a quality model (AIQM) [28,29] to assess AI-powered protection and AI [40] by identifying of important characteristics and subcharacteristics, as well as their metric-based assessment. In this study, AI means, applied to improve the cyber defence of ATS, are considered as a black box.possibilities of actors (regulators, developers, operators, customers) to choose countermeasures and provide acceptable risks [40]. The offered methodology favours the efficiency of the implemented countermeasures through the analysis of their consistency, costs, and so on.

The proposed methodology and technique allow for improving the trustworthiness and accuracy of the ATS safety assessment when AI components are used to protect cyber and physical assets, mainly by reducing the level of entropy through the decomposing factors of the impact of security threats and AI characteristics on safety. A detailed description of the principles is given in Section 2.2, Section 2.3 and Section 2.4.

### 2.2. Safety as a Top ATS Attribute: SIS Approach in the Context of Entropy

For ATS, safety is a top priority for the following reasons:human lives are at stake: autonomous transport systems have the potential to save thousands of lives by reducing the number of accidents caused by human error. However, if autonomous transport systems are not designed and tested to be safe, they can also pose a significant risk to human life.public perception: the success of autonomous transport systems depends on public acceptance. They will not be adopted if the public does not perceive them as safe. Safety is, therefore, crucial to building trust in autonomous transport systems.legal and regulatory requirements: autonomous transport systems are subject to legal and regulatory requirements that dictate safety standards. Failure to comply with these requirements can result in legal and financial consequences.reputation and liability: companies developing autonomous transport systems are vested in ensuring their vehicles are safe. If a severe accident were to occur, it could damage the reputation of the company and result in legal liability.technical challenges: developing safe autonomous transport systems is a complex technical challenge that requires significant research and development. Safety is, therefore, a top priority throughout the development process to ensure that the technology is reliable and robust enough to operate safely in a wide range of scenarios.

Traditional ATS are designed with a primary focus on safety and reliability requirements. Cybersecurity is usually added as an additional process that could make ATS vulnerable to various attacks. Modern ATS have a higher probability of cybersecurity vulnerabilities utilisation than hardware failures.

So, it is natural that safety should be considered primarily as a function of security, or more precisely, cybersecurity. Other attributes of reliability, repairability, etc., are also considered, but on the one hand, as mentioned, they have less impact on safety; on the other hand, their influence is studied in much more detail. Therefore, considering the approach called Security-Informed Safety (SIS) [35] or Cybersecurity-Informed Safety (CSIS) is relatively objective and appropriate for evaluating ATS safety and dependability. Within this paper, the authors do not distinguish between SIS and CSIS, because the scope of the application of SIS already considers cyberspace and cyberattacks.

The SIS/CSIS approach is entropy oriented since the impact of security on safety is characterised by tremendous uncertainty, which grows as physical assets are replaced by cyber assets at an increasing pace, as well as the diversification of cyber threats, alongside the increase in the intensity and impact of cyber wars. Considering the role played by ATSs, first of all, UAVs and UAV fleets in modern wars and their use in smart cities as efficient, robust, and dependable services [41], shows that the relevance of developing and applying the SIS-based approach appears very high.

### 2.3. SISMECA Technique

Security-informed safety means that security measures are integrated into safety systems to mitigate potential risks and threats. To perform a safety assessment of such systems, it is possible to perform FMECA to identify all possible failure modes and then classify their impact on safety, and in addition IMECA to identify potential threats and countermeasures. Such an approach requires a considerable number of resources and therefore is rarely achievable in practice.

In this work, we present the SISMECA technique aimed to focus on safety issues caused by either failures or threats. We describe FMECA and IMECA in the following subsections to provide a brief overview of existing approaches and then describe SISMECA and its specifics.

#### 2.3.1. FMECA

FMECA is a structured method for identifying, analysing, and prioritising potential failure modes and their effects on a system [42]. It is commonly used in engineering and manufacturing industries to identify potential failures and implement preventative measures.

For ATS, FMECA can be used to identify potential failure modes and their consequences and to develop strategies for mitigating these risks. By performing FMECA on ATS, experts can identify possible points of failure, assess the likelihood and severity of these failures, and prioritise them based on their criticality, so that the most critical risks are addressed first.

The FMECA process typically involves the following steps:system analysis: a thorough examination of the autonomous vehicle system is conducted to identify the various components and subsystems that make up the system;failure modes (Failure Mode, MOD) identification: potential failure modes for each component and subsystem are identified through brainstorming, previous experience, and research;failure modes analysis: each failure mode is analysed to determine its potential effects (Failure Effect, EFF) on the system and its criticality;criticality analysis: the criticality of each failure mode is assessed based on its severity (Failure Severity, SEV), frequency of occurrence (Failure Probability, PRO), and the likelihood of detection (Detectability, DET);risk prioritisation: the identified failure modes are prioritised based on their criticality (Risk Priority Number, RPN), and appropriate mitigation measures (Countermeasures, CTM) are implemented to address the highest-priority risks.

By using FMECA, experts can proactively identify potential failure modes and develop strategies for mitigating these risks before they occur. This helps ensure that autonomous transport systems are designed and built to the highest possible safety standards, essential for gaining public trust and acceptance of this new technology.

FMECA technique analysis is based on the development of a special table consisting of the elements MOD, EFF, SEV, PRO, DET, RPN, and CTM that are combined into a template.

#### 2.3.2. IMECA

IMECA is a technique intended for cybersecurity assessment considering refinements in the system. It can be applied to analyse the intrusions in the assessed object [25].

IMECA focuses on vulnerabilities that intrusions can utilise. In gap analysis, the detection of nonconformities and discrepancies (and related vulnerabilities in the case of cybersecurity assessment) can be implemented by separately identifying/analysing problems caused by human factors, techniques, and tools, considering the impact of the development environment. Then, after identifying all the vulnerabilities as a priority, it is possible to ensure the cybersecurity of critical instrumentation and control systems by implementing appropriate countermeasures.

Depending on the critical instrumentation and control system considered, each intrusion should be presented as a formal description that identifies any discrepancies (between “ideal”, i.e., described in the requirements, and real). Such a formal description should be made for the set of inconsistencies identified by a gap.

The IMECA table must represent each detected gap, and each discrepancy within the gap can be represented by a row in this table, considering the characteristics of the product and/or process feature. A separate table for each gap contains the vulnerabilities identified during the gap analysis. All individual tables are then combined into a common IMECA table.

The IMECA technique for cybersecurity analysis is based on the development of a special table considering the following elements:threats to system operation (THR);vulnerabilities of system components and operation processes (VLN);attacks on system assets (ATA);effects for system operation (EFF);assessment of the criticality of effects (CRT) estimated by the probability of an event and corresponding effects of the attacks (PRE), as well as the severity of the effects (SVE):
CRT = PRE × SVE.

A set of sub-stories (SS1, …, SSn) is formed as a list of analysed events related to different attacks on vulnerabilities. Sub-stories detail elements of stories and are separate rows of the IMECA table (see Table 3).

In [40], it was shown that a table could be added by columns of recovery (REC) as a metric of criticality and countermeasures (CTM).

REC is a level of system operation maintainability regarding the possibility of recovery after an attack on assets. The lower the REC value, the lower the time and cost of recovery. In this case:CRT = PRE × SVE × REC.

Table 4 provides a template of the IMECA table extended with the REC and CTM columns.

Furthermore, the IMECA table could be extended by specifying actors (ACT) implementing CTMs (see Table 5).

Using IMECA, a matrix of cyber risks of successful intrusions can be developed accordingly (Figure 2a). The green colour marks a square of low risk (sub-stories SSi, SSj, SSk), the yellow colour marks a square of moderate risks (SSl, SSp, SSq), and the red colour marks a square of high or unacceptable risk (SSr, SSs, SSt).

The IMECA process typically includes the following steps:Determination (collection) of stories and sub-stories set for the analysed system considering the domain experience;Decomposition and analysis of sub-stories, filling in the IMECA table and the matrix of risks;Analysis of the matrix or cube of criticality and assessment of the risk value;Determination of a set of countermeasures considering the responsibility of actors;Specification of criteria for the choice of countermeasures;Choice of countermeasures and verification of acceptable risk.

IMECA fits well for cybersecurity assessment, but it does not separate effects caused by attacks on vulnerabilities to safety-related and non-safety-related ones.

#### 2.3.3. SISMECA Ontology and Template for Assessment

In [39], XMEA was suggested as a unified safety and security assessment approach. The main idea was to utilise the same universal approach for different domains. The disadvantage of such an approach lies in the necessity of carrying out separate procedures for safety and security, which, though they follow the same sequence of operations, still are significantly resource consuming.

In SISMECA, the ‘Effects’ column is divided into two columns: one with an effect on security (EFF_SEC) and one with an effect on safety (EFF_SAF); a template of such a table is shown in Table 6. This allows for focusing only on security issues that affect safety, implementing the SIS approach.

Figure 3 shows the SISMECA ontology model, providing its relations with known techniques such as FMEA, FMECA, and IMECA.

Extending FMEA by analysing criticality, FMECA is obtained; while focusing on diagnostics, the result is FMEDA, and so on. SISMECA is obtained from IMECA by profiling safety-related rows.

Considering the SISMECA ontology model and template, Figure 4 illustrates the relations among different safety, security, and reliability risk assessment techniques. SISMECA considers security-related risks for safety, and SafMECA combines the results of the SISMECA and FMECA analysis.

### 2.4. SISMECA-Based Safety Assessment and Ensuring

SISMECA is suggested to be used as a key method in AI quality model-based strategy. An overview of this approach is shown in Figure 5 and includes the following principles [40]:development of scenarios set describing the operation of ATS under cyberattacks and actor activities considering AI contribution to system protection for the analysed domains;scenario-based development and analysis of user stories describing different cyberattacks, their influence, and ways to protect ATS from them via AI means/platforms;profiling of AI platform requirements by use of characteristics-based AI quality models;SISMECA-based assessment of cyberattack criticality and effect on safety, as well as efficiency of countermeasures that actors can implement.

It should be mentioned that the application of artificial intelligence as a means to protect ATS assets in the context of the SIS approach is considered as follows:AI means can reduce, but not completely eliminate, the risks of dangerous ATS failures caused by cyberattacks;failures or other anomalies of AI means can cause dangerous states of ATSs.

## 3. Security-Informed Safety Assessment of ATSs

Based on the developed SIS methodology of analysis and use of the AI quality model, further in this section, examples of its application for various types of ATS are provided based on consideration of specific user stories: US-ATS.A (User Story for Autonomous Transport Systems from Aviation domain), US-ATS.M (User Story for Autonomous Transport Systems from Maritime domain), US-ATS.S (User Story for Autonomous Transport Systems from Satellite domain). In these systems, AI tools are used to protect cyber assets in the face of information intrusions, which can also be supported by AI tools.

Analysing user stories, as well as features of relevant AI systems and tools, a quality model of the investigated system (as a subset of the general model) is built with the help of a general AI quality model. The purpose of building such an AI model from the point of view of the SIS approach for the system under study is as follows:determine the need to consider the safety issue in this model from the point of view of the effects of failures and any anomalies of AI tools on the ATS safety.based on the built AI model, determine which of its characteristics affect safety-related risks. As part of this work, only the impacts related to security (cybersecurity) are analysed, since the SIS approach is being investigated. Analysis of the impact of other AI characteristics on safety is a separate task.AI tools within the framework of the quality model and follow-up assessment using SISMECA tables can and should also be analysed from the point of view of the presence of their vulnerabilities and their impact on the ATS safety.

The aspect of AI-powered attacks and AI-powered protection is considered through a set of competitive scenarios described in [43] by

expanding the number of cases of cyberattacks, considering their reinforcement using AI, which will lead to an increase in the number of rows of IMECA/SISMECA tables and the corresponding tracing of analysis results (modes, effects, and criticality).increasing opportunities for asset protection (countermeasures) thanks to the use of AI (AI-powered protection).

It should be mentioned that AI can also be considered as an object of protection when it is a subsystem of the ATS performing its respective functions. In this case, its vulnerabilities should be considered, as well as all subsystems.

Thus, the structure of the analysis of AI-based ATS cases using the SIS approach is the following:analysing the corresponding user story from the point of view of security and safety issues and the role of AI means.profiling of AI quality model based on the general one.carrying out IMECA and SISMECA analysis and developing relevant tables and criticality matrixes.analysing acceptable risks after the application of countermeasures.

To demonstrate the examples of the proposed methodology application for the three domains (maritime, aviation, space), the following structure of the further Section 3.1, Section 3.2 and Section 3.3 is applied. The first-level Section 3.1.1, Section 3.2.1 and Section 3.3.1 contain the description of the user story from the respective domain (maritime, aviation, space). The mentioned user story can be used during further requirement elicitation for the cybersecurity of ATS in the particular domain, where AI is applied and cybersecurity restrictions are to be applied. The second-level Section 3.1.2, Section 3.2.2 and Section 3.3.2 describe the profiling of the user story via building the quality model of AI tools and AI systems mentioned in the user story. The third-level Section 3.1.3, Section 3.2.3 and Section 3.3.3 contain a cybersecurity assessment of the user story via the application of IMECA with the indication of the initial risks in the risk matrix, countermeasures implemented by four stakeholders (developer, regulator, operator, user), and the respective changes in the risk matrices. Finally, Section 3.1.4, Section 3.2.4 and Section 3.3.4 describe SISMECA safety assessment for the user story from the respective domains. The goal here is to demonstrate how cybersecurity reflects on functional safety. Cybersecurity violation impact (cyberattacks on vulnerabilities) is analysed in its application to the corresponding safety impact.

### 3.1. Case Study for the Maritime Domain

#### 3.1.1. User Story US-ATS.M: Human Machine Interface of the Shore Control Centre

Experience with using an autonomous ship revealed that the Human–Machine Interface (HMI) of the Shore Control Centre (SCC) is vulnerable to AI-powered spoofing, tampering, information disclosure, denial of service, and elevation of privilege attacks. Through the HMI, humans can operate the ship under various conditions. As for spoofing, an attacker could access the system and critical information. This will influence the entire infrastructure and cause a bad reputation for the company or even litigation. As for tampering, data tampering in this system will put the ship in danger since, through this system, unauthorised humans on the shore can control and monitor the ship. As for information disclosure, the HMI contains information crucial for the ship’s sailing. Disclosure of this information could lead to damage since it relates to the vessel’s navigation and management. As for denial of service, availability is critical for secure sailing. If this system becomes unavailable, the ship will be control-less and invisible to the SCC. As for the elevation of privilege, an attacker with administrative rights to the system could access sensitive data about the vessel’s condition, customers, and passengers. This could raise legal issues for the shipping company. To respond effectively to cyberattacks, the developer proposed an AI-based Intrusion Detection System (IDS), using a Recurrent Neural Network (RNN). The proposed methodology takes either of the two intrusion detection benchmark datasets and pre-processes it. The pre-processed data are then given to the proposed model for training, and once the model is trained, the weights are saved for future use. The test data are then evaluated on the trained model for intrusion classification. Two benchmark datasets were used to evaluate the proposed model: NSL-KDD and CICIDS. NSL-KDD is an improved version of the famous KDD cup 99 dataset and contains around 41 features, and the class labels are labelled either normal or specific types of attack. CICIDS is a relatively new dataset that is being used as a benchmark for intrusion detection systems. One of the reasons for choosing this dataset is that it contains wider data for the most recent and more common attacks over any other dataset.

#### 3.1.2. Profiling of the User Story US-ATS.M

A quality model for the AI-based IDS system described in the user story US-ATS.M as an AI system is shown as a subgraph with the characteristics of AIS, being important for the analysed system; it is marked with a grey colour. For this system, the following characteristics should be taken into consideration:for the first level of AI quality: LFL (Lawfulness), EXP (Explainability), and TST (Trustworthiness). In this case, the characteristic ETH (Ethics) is not obvious.for the second level of AI quality: CMT (Completeness), CMH (Comprehensibility), TRP (Transparency), INP (Interpretability), INR (Interactivity), and VFB (Verifiability) for AI explainability; RSL (Resiliency), RBS (Robustness), SFT (Safety), SCR (security), and ACR (Accuracy) for AI trustworthiness.for the first level of AIP quality: ADT (Auditability), AVL (Availability), EFS (Effectiveness), RLB (Reliability), MNT (Maintainability), and USB (Usability).

VFB, DVS, RSL, RBS, SFT, SCR, and ACR characteristics are general for AI and AIP quality sub-models (AIG). To show this on the graph (Figure 6), the vertices VFB, DVS, RSL, RBS, SFT, SCR, ACR, and AIP are connected to the vertex AIG using dashed lines. Profiled quality model (Figure 6) can be applied to assess the AIS using metric-based techniques [27,30]. This technique comprises the following operations: assessing metrics values for AI quality characteristics/sub-characteristics, determining their weights, and calculating integrated quality indicator using additive convolution.

#### 3.1.3. IMECA Cybersecurity Assessment for the User Story US-ATS.M

To demonstrate the utility of the IMECA technique for the maritime domain ATSs, we analysed Story US-ATS.M. Results of the application of the IMECA technique are shown in Table 7. Criticality level was assessed for two cases: with the application of recovery procedures and without application. Moreover, a feature of this table is that the column of countermeasures is divided into four columns, presenting results of possible decreasing of criticality for the corresponding sub-stories considering CTM implemented by four actors: developer (DEV), regulator (REG), operator (OPR), and the user (USR).

The results of the SSs analysis (matrixes of risks) are shown in Figure 7 and Figure 8. The first matrix did not consider the application of countermeasures by different actors (Figure 7). The matrixes (Figure 8) describe a potential risk decrease after implementing countermeasures by developers, regulators, and operators.

The application of countermeasures for this example shows that the efforts of DEV, REG, and USR can achieve the same effect, namely, reducing the risks of successful attacks for states SS1-SS5 due to decreasing their probabilities. OPR’s efforts for this case cannot reduce the risks.

#### 3.1.4. SISMECA Safety Assessment for the User Story US-ATS.M

Considering the SISMECA template introduced in Section 2.3.3. (Table 7), we analysed security violation impact (cyberattacks on vulnerabilities) for corresponding safety impact. One row was added to the SISMECA table in order to consider possible countermeasures to reduce ATS critical failures.

In addition, Table 8 analyses the effects in terms of safety, as well as capabilities of reducing the effects of cyberattacks on such failures, provided that countermeasures are proposed and implemented only at the developer level:security ensuring means (information technologies level, CTMsec);safety ensuring means (information and operation technologies level CTMsaf);combined usage of ensuring means at IT and OT levels.

The criticality matrix (Figure 9) serves to illustrate an effect from the implementation of the means mentioned above, in terms of initial high criticality level for substory (M × H) to medium risk levels SS1sec (L × H), SS1saf (M × M) and low-risk level SS1comb (L × M) in the case of combined usage.

### 3.2. Case Study for the Aviation Domain

#### 3.2.1. User Story US-ATS.A: UAV-Based Surveillance System

A company used UAVs as components of an aerial surveillance system. Experience with the utilisation of UAVs revealed that UAVs are vulnerable to AI-powered GPS spoofing attacks and AI-powered GPS jamming attacks. These attacks can happen because of weaknesses in the navigation system. GPS spoofing occurs when someone uses a radio transmitter to send a counterfeit GPS signal to a receiver antenna to counter a legitimate GPS satellite signal. Most navigation systems are designed to use the strongest GPS signal, and the fake signal overrides the weaker but legitimate satellite signal. GPS jamming happens when an attacker blocks GPS signals altogether. An AI-powered GPS spoofing attack is initiated when the UAV uses the autopilot flight mode, which can be automatically initiated when the drone has lost contact with the ground control station (GCS). This kind of connection disruption is triggered by a jamming attack (jamming attacks forcibly switch the drone’s flight mode into autopilot even if the drone is still within the communication range of the control unit) from the attacker’s ground station. After hacking the UAV using AI-powered jamming attacks, the attacker forces the UAV to land in a pre-picked zone by spoofing GPS signals. In such a situation, the UAV cannot initiate its safe return-to-home (RTH) facility due to the absence of connection with the GCS, as in typical challenging scenarios. As countermeasures, a deep-learning-based, adaptive Intrusion Detection System (IDS) was used by an aerial surveillance system developer for a UAV to identify its intruders and ensure its safe RTH. In the proposed IDS, Self-Taught Learning (STL) with a multiclass SVM (Support Vector Machine) was used to maintain the high true positive rate of the IDS, even in uncharted territory. A self-healing method in the IDS recovery phase uses the Deep-Q Network, a deep reinforcement learning algorithm for dynamic route learning to facilitate the UAV’s safe RTH.

#### 3.2.2. Profiling of the User Story US-ATS.A

A quality model for the AI-based IDS system, which is described in the user story US-ATS.A as an AI system, is shown as a subgraph with the characteristics of AIS, which are important for the analysed system and marked with grey colour. For this system, the following characteristics should be considered:for the first level of AI quality: ETH, LFL, EXP, and TST (in this case, all characteristics of the first level of AI quality are essential for this AIS);for the second level of AI quality: FRN for AI ethics; CMT, CMH, TRP, and VFB for AI explainability; RSL, RBS, SFT, SCR, and ACR for AI trustworthiness;for the first level of AIP quality: ADT, AVL, CNT, EFS, and RLB.

The characteristics VFB, RSL, RBS, SFT, SCR, and ACR are general for AI and AIP quality sub-models. A profiled quality model (Figure 10) can be applied to assess the AIS using metric-based techniques [30]. This technique comprises the following operations: assessing metrics values for AI quality characteristics/sub-characteristics, determining their weights, and calculating the integrated quality indicator using additive convolution.

#### 3.2.3. IMECA Cybersecurity Assessment for the User Story US-ATS.A

Results of the IMECA technique application are presented in Table 9. Criticality level was assessed for two cases: with the application of recovery procedures and without application. Moreover, a feature of this table is that the column of countermeasures is divided into four columns, presenting results of possible decreasing of criticality for the corresponding sub-stories considering CTM implemented by four actors (DEV, REG, OPR, USR).

The results of SSs analysis (matrixes of risks) are shown in Figure 11 and Figure 12. The first matrix did not consider the application of countermeasures by different actors (Figure 11). The matrixes (Figure 12) describe a potential decrease in risks after implementing countermeasures by developers, regulators, and operators.

The application of countermeasures for this example can be efficient due to efforts of DEV, REG, and USR and achieve the same effect of reducing the risks of successful attacks for states SS1 and SS2. OPR’s efforts in this case cannot reduce the risks.

#### 3.2.4. SISMECA Safety Assessment for the User Story US-ATS.A

Results of SISMECA, performed in the same way as in Section 3.1.4, are provided in Table 10 and Figure 13.

### 3.3. Case Study for the Space Domain

#### 3.3.1. User Story US-ATS.S: Smart Satellite Network

A company used a Smart Satellite Network (SSN), which can be defined as a network of satellite systems, the processes of which have been augmented by including smart sensors and actuators. Experience with the utilisation of the SSN revealed that it is vulnerable to the following AI-powered attacks: Distributed Denial of Service (DDoS) attacks because of weaknesses in communication protocol or power supply devices (A1), targeted attacks because of deficiencies in power supply devices (A2), tampering attack because of the Operating System (OS)/firmware weaknesses (A3), Man-In-The-Middle (MITM) (A4) attacks, and data manipulation attacks (A5) because of weaknesses in channels responsible for transmitting data between satellites and ground stations. It is vital to note that A1 resulted in increased network interactions, A1 and A2 resulted in power depletion, A3 resulted in a non-functional state known as bricks, and A4 and A5 resulted in violating the confidentiality/integrity of smart satellite data. To detect and investigate the cyberattacks, the developer proposed a DL-based network forensic framework, consisting of augmented satellites and IoT devices, called INSAT-DLNF (Intelligent Satellite Deep Learning-based Network Forensics). For the development of the proposed INSAT-DLNF framework, the Gated Recursive Unit Recurrent Neural Network (GRU-RNN) was utilised, which made it possible to create a model for detecting attacks in collected network traces. Big data collections, such as NSL-KDD, UNSW-NB15, and Bot-IoT, were analysed for investigating attack events, and their traces were analysed for developing reliable network forensics models.

#### 3.3.2. Profiling of the User Story US-ATS.S

A quality model for the AI-based IDS system described in the user story US-ATS.S as an AI system is shown as a subgraph with the characteristics of AIS, which are important for the analysed system and marked with grey colour. For this system, the following characteristics should be taken into consideration:for the first level of AI quality: LFL, EXP, and TST (in this case, characteristic ETH is not obvious);for the second level of AI quality: CMT, TRP, INR, and VFB for AI explainability; DVS, RSL, SFT, and SCR for AI trustworthiness;for the first level of AIP quality: ADT, AVL, EFS, RLB, MNT, and USB.

VFB, DVS, RSL, and CSR characteristics are general for AI and AIP quality sub-models. Profiled quality model (Figure 14) can be applied to assess the AIS using metric-based techniques [30]. This technique comprises the following operations: assessing metrics values for AI quality characteristics/sub-characteristics, determining their weights, and calculating the integrated quality indicator using additive convolution.

#### 3.3.3. IMECA Cybersecurity Assessment for the User Story US-ATS.S

To demonstrate the utility of the IMECA technique for the space ATSs, we analysed the user story US-ATS.S. Results of the IMECA technique application are presented in Table 11. Criticality level was assessed for two cases: with and without the application of recovery procedures. Moreover, a feature of this table is that the column of countermeasures is divided into four columns, presenting results of possible decreasing of criticality for the corresponding sub-stories considering CTM implemented by four actors (DEV, REG, OPR, USR).

The results of SSs analysis (matrixes of risks) are shown in Figure 15 and Figure 16. The first matrix did not consider the application of countermeasures by different actors (Figure 15). The matrixes (Figure 16) describe a potential decrease in risks after implementing countermeasures by developers, regulators, and operators.

The application of countermeasures for this example can be efficient due to efforts of DEV, REG, and USR and achieve the same effect of reducing the risks of successful attacks for states SS1, SS4, and SS5. OPR’s efforts for this case cannot reduce the risks.

#### 3.3.4. SISMECA Safety Assessment for the User Story US-ATS.I

Results of SISMECA, performed in the same way as in Section 3.1.4., are provided in Table 12 and Figure 17.

## 4. Towards the Roadmap of AI Cybersecurity for ATS in the Context of the SIS Approach

Based on the analysed information and the cybersecurity concerns connected with the application of AI in ATS, we developed an innovative technology roadmap entitled “AI/ML Cybersecurity for Aviation/Space and Maritime Autonomous Transport” [43] within the project ECHO (the European network of Cybersecurity centres and competence Hub for innovation and Operations) funded under the H2020 programme.

Elements of the developed evaluation methodology, in particular AIQM and their profiling for specific types of ATS, IMECA-based analysis of cybersecurity considering the use of AI, and a review of almost 30 user stories for three types of ATS (maritime, aviation, space) were included into the roadmap developed in the frame of the project ECHO [40]. As far as the main concern of the roadmap was exactly in ATS cybersecurity (without safety consideration) and the need to investigate the research results of the impact of ATS cybersecurity on functional security, we see the need to clarify some positions of the developed roadmap. This section describes the possibilities for further extension of the roadmap, considering the aspect of safety and the proposed SIS approach.

### 4.1. AI Cybersecurity for the ATS Roadmap Evolution

The following high-level statements provide a general framework for enhancing the roadmap for the next several years. The exact timeframe was challenging to indicate precisely due to the apparent complexity and constant growth of AI evolution velocity. As the most realistic in terms of the possible prognosis, the period of 3 years has been chosen considering the rising application of generative AI and the accompanied challenges (some of which cannot even be known now, which is a subject of raising concern in society). It is essential to adapt and refine the roadmap and the statements based on the specific needs, advancements, and challenges within each domain of ATS. By considering the evolution of cybersecurity tools, AI as a whole, and generative AI particularly, we propose the refining the roadmap with the following points of focus:Constantly reassess the current threat landscape: conduct an ongoing and comprehensive assessment of the current cybersecurity threats facing ATS, considering various domains (maritime, aviation, space, ground). Identify potential vulnerabilities and attack vectors specific to AI-powered ATS.Implement advanced threat detection: deploy AI-powered threat detection systems that leverage machine learning algorithms, including behaviour-based analysis and anomaly detection. Continuously update and enhance these systems to adapt to evolving cyber threats.Strengthen AI governance: develop robust AI governance frameworks that ensure the ethical, secure, and safe development, deployment, and operation of AI systems in ATS. Establish accountability, transparency, and monitoring mechanisms to mitigate cybersecurity risks and functional safety threats.Enhance intrusion detection and prevention: continuously improve intrusion detection and prevention capabilities by integrating advanced cybersecurity tools with AI algorithms. Leverage AI techniques, such as deep learning, for the more accurate and proactive identification of cyber threats in real time.Foster collaboration for threat intelligence: establish partnerships and alliances with industry stakeholders, cybersecurity experts, and academia to share threat intelligence and best practices. Leverage collective knowledge to enhance the overall cybersecurity and safety posture of ATS.Implement secure data sharing: develop the frameworks that ensure the confidentiality, integrity, privacy, and cybersafety (attribute connecting cybersecurity with safety) of data exchanged within and between ATS. Utilise encryption, access controls, and blockchain technology for secure and trusted data sharing.Embrace generative AI security and cybersafety: address the potential cybersecurity risks and safety threats associated with generative AI algorithms in ATS. Develop robust security testing and validation procedures specific to generative AI models to identify and mitigate vulnerabilities.Continuously monitor and update security measures: establish a continuous monitoring system to track the effectiveness of cybersecurity measures in AI-powered ATS. Regularly update security protocols, leverage threat intelligence, and adapt to emerging cyber threats.Foster AI-driven threat hunting: utilise AI-driven threat hunting techniques to proactively search for potential cyber threats in ATS. Develop AI models to analyse large datasets, identify patterns, and detect anomalies to enhance proactive defence strategies.

### 4.2. Evolution of Cybersecurity Tools, AI as a Whole, and Generative AI in Particular

The evolution of cybersecurity tools, AI as a whole, and generative AI can significantly impact and shape the roadmap for AI cybersecurity in ATS. Here is how these factors can influence the roadmap:Evolution of cybersecurity tools:Advanced threat detection: as cybersecurity tools evolve, they may incorporate more advanced techniques such as behaviour-based analysis, anomaly detection, and machine learning algorithms. This can enhance the effectiveness of threat detection in ATS, enabling earlier detection and response to cyber threats, thus decreasing the risk of safety violation.Real-time monitoring and response: advanced cybersecurity tools may provide real-time monitoring capabilities, allowing immediate response to potential cyberattacks. This can enable faster incident containment and mitigation, reducing the impact on the safety and functionality of ATS.Integration with AI systems: integrating cybersecurity tools with AI systems can result in more intelligent and adaptive defence. AI algorithms can analyse vast amounts of data to identify patterns, detect anomalies, and respond to emerging threats in real-time, enhancing the overall cybersecurity and, at the end of the day, safety posture of ATS.Evolution of AI as a whole:Improved threat intelligence: AI algorithms can analyse large datasets and identify patterns humans might overlook. This can improve threat intelligence, allowing for more accurate identification and prediction of cyber threats in ATS, which can influence safety.Advanced intrusion detection and prevention: AI-powered intrusion detection systems can continuously learn and adapt to evolving cyber threats. They can detect and prevent sophisticated attacks that traditional rule-based systems might miss, bolstering the cybersecurity defences of autonomous transport systems.Enhanced authentication and access control: AI can facilitate advanced authentication mechanisms, such as biometrics and behavioural analysis, to strengthen access control in autonomous transport systems. This can minimise the risk of unauthorised access and mitigate potential cyberattacks.Influence of generative AI:Potential cybersecurity risks: using generative AI algorithms in ATS introduces potential cybersecurity risks. These algorithms can be vulnerable to adversarial attacks or manipulation, leading to safety risks. The roadmap needs to account for developing robust defences against such attacks.Security testing and validation: generative AI algorithms require thorough security testing and validation to ensure their integrity and attack resilience. The roadmap should incorporate processes for testing and validating generative AI models to identify potential vulnerabilities which can target safety threats and address them before deployment.Robust AI governance: the roadmap should consider robust AI governance frameworks that ensure the ethical, secure, and safe development, deployment, and operation of generative AI algorithms in ATS. This provides accountability, transparency, and continuous monitoring of AI systems to mitigate cybersecurity risks.

## 5. Discussion

The following section elaborates on the limitations of the used XMECA-based techniques in the research and the ways of further improvements towards the regulation of AI and cybersecurity in the autonomous transport systems domain in EU, as well as considering raising threats to the environment in the context of Chemical, Biological, Radiological, Nuclear, and Explosives (CBRNe).

### 5.1. Interconnection between OT and IT

OT (Operational Technology) and IT (Information Technology) are interconnected domains in the context of functional safety and cybersecurity analysis of ATS where AI is used. OT refers to the technology and systems used to monitor and control physical processes and operations in the ATS, such as sensors, actuators, controllers, and other devices. IT refers to the technology and systems used to manage and process data and information in the system, such as servers, networks, databases, and software applications. In the context of functional safety and cybersecurity analysis of ATS, the interconnection of OT and IT is significant because it creates potential entry points for cyber threats that could compromise the safety and security of the system. For example, an attacker may use a cyberattack to compromise a server or network in the IT domain (e.g., via attacking the AI), which could then allow them to gain access to the OT domain and compromise the sensors or controllers that are used to monitor and control physical processes of ATS. Considering both the OT and IT domains, it is important to holistically integrate functional safety and cybersecurity analysis to address this potential vulnerability.

The proposed methodology tries to cover the gap by addressing this connection holistically, thus ensuring that the system is safe and secure and that potential cyber threats are effectively mitigated.

To reach this goal, the methodology encompasses actions for

identifying and assessing the risks and vulnerabilities associated with both OT and IT domains. This includes evaluating the potential impact of cyber threats on the safety and reliability of the autonomous transport system.implementing appropriate safety and security controls in both OT and IT domains. This may include implementing physical, technical, and procedural safeguards to prevent or mitigate the effects of cyber threats.establishing clear communication and collaboration between OT and IT teams. This includes sharing information about potential risks and vulnerabilities and ensuring that safety and security controls are implemented and maintained consistently across both domains.incorporating AI into the functional safety and cybersecurity analysis process. AI can identify potential threats and vulnerabilities in the system and develop and implement effective safety and security controls.decomposition of reasons related to cybersecurity that affect safety risks, significantly reducing the uncertainty of system behaviour and the consequences of cyberattacks on the vulnerability of ATSs. This is due to the possibility of careful analysis of the chain “threat–vulnerability–attack–consequences–countermeasures” from the point of view of safety. The accuracy of determining the effect of the use of AI, as well as the consequences of cyberattacks on the components of ATSs, which are implemented using AI, is increased thanks to the decomposition of the set of characteristics of AI using a quality model.

### 5.2. Application of Hardware-Oriented Techniques Which Were Developed for Functional Safety Analysis to Software-Oriented Cybersecurity Analysis

Initially developed as hardware-oriented analysis, the techniques previously described and adopted in many research papers are methods used to ensure the safety and cybersecurity of complex hardware and software systems, such as ATS. These techniques were initially developed to ensure that these systems were safe to operate and would not fail. However, as AI-based ATS systems have become more complex and interconnected, the need to protect them from cybersecurity threats has become increasingly important.

Fault injection testing can be mentioned among the examples of a hardware-oriented analysis technique applied to cybersecurity. This technique intentionally introduces faults into a system to see how it responds. In the context of cybersecurity, fault injection testing can be used to simulate attacks and determine how a system will react to them. This can help identify vulnerabilities in the system that attackers could exploit. Another example of a hardware-oriented technique used for the system design to improve its cybersecurity is redundancy. Redundancy involves adding extra components or subsystems to a system to ensure it can continue functioning even if one or more components fail. In the context of cybersecurity, redundancy can be used to ensure that critical systems can continue to work, even if attackers target them.

Pros of using hardware-oriented techniques for analysis and design to improve cybersecurity are as follows:They have been used successfully for safety and can be adapted for cybersecurity because of the experience and gained lessons learned during application in safety-critical domains;They can be very effective at identifying vulnerabilities and threats in systems;They can help ensure that critical systems continue to function even if attackers target them;They can help build resilience into systems, making them more attack resistant;Cons of using hardware-oriented techniques for cybersecurity are as follows:They can be expensive and time-consuming to implement;They may not be effective against all types of attacks;They may be less effective against attacks specifically designed to bypass them;They may not be appropriate for all types of systems.

It is important to note that while hardware-oriented techniques used for analysis and design can be very effective at identifying vulnerabilities and building resilience into systems, they are not a silver bullet for cybersecurity. It is essential to use a combination of techniques, including hardware-oriented and software-oriented approaches, to ensure the cybersecurity of ATS.

### 5.3. EU Initiative on AI: Ensuring the Appropriate Safety and Liability Regulations

The emergence of AI, in particular the complex enabling ecosystem and the feature of autonomous decision making, requires a reflection on the suitability of established rules on safety and civil law questions on liability [44]. For instance, advanced robots and IoT products empowered by AI may act in ways not envisaged when the system was first implemented. Given AI’s widespread use, horizontal and inter-sectoral rules may need to be reviewed.

The EU Strategic Framework on Health and Safety at Work 2021–2027 [45] already addresses the intended use and foreseeable use or misuse of products when placed on the market, e.g., Machinery Directive [46]; Radio Equipment Directive [47]; and EU’s General Product Safety Regulation [48], which is agreed and going to be updated in the nearest upcoming months [49], as well as other specific safety rules, among others. This has led to developing standards and regulations in AI-enabled devices continuously adapted to technological progress.

The further development and promotion of such safety standards and support in EU and international standardisation organisations will help European businesses benefit from a competitive advantage and increase overall trust (consumers, stakeholders, etc.) via a combination of cybersecurity, safety, and SIS-based methodology. These regulations should also cover interoperability, cross-domain, and inter-sectoral issues, which are crucial for offering stakeholders meaningful choices and ensuring fair competition.

### 5.4. Application of AI-Powered Cyberattacks: CBRNe Issues

The probability of a successful cyberattack by hackers of various levels (starting from hacktivists and finishing with groups funded by influential states that have been recognised as terrorists or commit terrorist acts of aggression) using AI on autonomous vehicles will highly likely lead to dramatic consequences for functional safety at the infrastructure level. The current status of the development of AI and autonomous vehicles of various fields of application (on-ground, maritime, aviation, and space), as well as methods of conducting a hybrid war (including cyberwarfare and cyberterrorism), allow unnamed groups to violate not only the functional safety of individual autonomous vehicles but also intercept them, influencing them in such a way as to use them to damage critical infrastructure facilities.

Cyberattacks on critical infrastructure objects and facilities (e.g., oil and gas complex facilities and pipelines, power generation and transmission facilities, fibre-optic communication and data transmission lines, heat and water supply and sewerage treatment, waste treatment, chemical processing plants) that affect the quality of life of a large number of civilians can entail not only a one-time deterioration in the quality of life (e.g., security, human health, economic development) of the population in a particular territory, but they also carry epidemiological risks of various severity and criticality levels, which were even not faced by humanity before. In this context, the importance and necessity of considering such attacks are rapidly growing, taking into account the aspects of CBRNe (National Strategy for Chemical, Biological, Radiological, Nuclear, and Explosives) [50,51]. These are types of hazards that pose a significant threat to critical infrastructure and public safety. AI-powered cyberattacks on CBRNe-related systems can have devastating consequences, such as releasing toxic substances or disrupting emergency response services. AI-powered cyberattacks on CBRNe-related systems may become more targeted and sophisticated in the next five years. For instance, attackers may use AI-powered social engineering tactics to trick employees or emergency responders into providing access to sensitive systems or information. They may also use AI-powered malware or ransomware to disrupt operations or hold critical systems hostage. Investing in robust cybersecurity measures, such as secure network design, access control, and threat detection and response, is essential to counter these threats. It is also crucial to ensure that emergency response teams can access reliable communication networks and information systems protected against cyber threats. Additionally, continued research and development in AI-powered cybersecurity technologies will be necessary to stay ahead of emerging threats, e.g., AI-powered threat detection systems that can identify and respond to CBRNe-related cyber-informed safety threats in real time could be critical in preventing catastrophic events.

### 5.5. Challenges of Security-Informed Safety Analysis of ATSs

The Security-Informed Safety Analysis of ATS presents several challenges due to the complex nature of these systems and the criticality of ensuring their safety and security. Here are some of the key challenges associated with such analysis for ATSs:Multiple interconnected components, such as sensors, actuators, control systems, and communication networks, are used in ATSs. It can be difficult to analyse the safety and security implications of such complex systems since flaws or vulnerabilities in one component might have cascading effects on the system as a whole.ATSs operate in dynamic and unpredictable environments, interacting with other transportation systems and infrastructure. Assessing the safety and security of these systems requires considering various scenarios and potential risks associated with different operational conditions, such as adverse weather or unexpected events.As autonomous transport technology evolves rapidly, there is a lack of standardised frameworks, guidelines, and regulations specifically addressing the safety and security aspects of these systems. This creates challenges in conducting comprehensive analysis, as there is no universally accepted methodology or set of criteria to evaluate the safety and security of ATSs.ATSs often involve human interaction, such as operators or maintenance personnel. Considering human factors, such as user behaviour, training, and response to system failures, is essential for comprehensive analysis. However, analysing and incorporating these factors into the analysis can be complex and requires a multidisciplinary approach.

Thus, addressing these challenges requires a collaborative effort involving experts from various domains, including engineering, cybersecurity, transportation, and policymaking. It is important to continuously improve and refine security-informed safety analysis methodologies, frameworks, and regulations to ensure the safe and secure deployment of ATSs in real-world scenarios.

## 6. Conclusions

The SIS-based methodology and the SISMECA technique, in combination with the well-known FMECA technique, provide a comprehensive safety assessment of ATSs and other safety-critical systems due to a combined analysis of the risk due to violations and damages of physical and cyber assets. SISMECA is an example of an attribute-scalable analysis technique, since it allows one to assess ATS characteristics under various options of artificial intelligence applications, such as AI-powered protection against AI-powered attacks [52].

As AI-based cybersecurity tools evolve, so do the functional safety and cybersecurity concerns surrounding their use in attacks on ATS. Various factors, including the increasing complexity of ATS, the growing sophistication of cyber attackers, and the need for improved safety and security in these systems, drive the evolving nature of these concerns.

One key concern is the potential for AI-based cybersecurity tools to be used in cyberattacks on ATS. While these tools can effectively detect and respond to cyber threats, they can also be used to launch attacks. For example, an attacker could use an AI-based cybersecurity tool to scan a system for vulnerabilities and then exploit those vulnerabilities to gain unauthorised access or cause damage to the system. Another concern is the potential impact of cyberattacks on the safety and functionality of ATS. These systems rely on a range of sensors, processors, and controllers to operate, and a cyberattack on any of these components could have significant safety implications. For example, an attacker could compromise the sensors an autonomous vehicle uses, causing it to misinterpret its surroundings and potentially cause an accident.

To address these concerns, a growing focus is on co-engineering functional safety and cybersecurity in developing AI-based cybersecurity tools for ATS. The co-engineering of functional safety and cybersecurity for cybersecurity assurance of AI-based ATS is an ongoing process that involves collaboration between safety, cybersecurity, AI, and ATS experts and integrating functional safety and security considerations throughout the entire design and development process, ensuring that both functional safety and cybersecurity requirements are met at every stage of the development process. It also involves monitoring and maintaining these systems to detect and respond to potential safety hazards and security threats. This collaboration is essential to ensure that the system is both safe and secure and that potential cyber threats are effectively mitigated. As AI-based ATS and cybersecurity tools used for attacks and protection continue to evolve, it is important to continue to incorporate functional safety and cybersecurity co-engineering principles to ensure that these tools are effective at defending against cyber threats in ATS.

The investigated approach of combining SIS-, IMECA-, and AIQM-based assessment techniques serves several important purposes:Addressing emerging challenges: The integration of AI in autonomous transport systems introduces new challenges and risks, particularly in terms of cybersecurity. By adopting this approach, the investigation aims to proactively address these emerging challenges and develop methodologies and techniques that consider the application of AI in the context of cybersecurity and safety.Holistic assessment: The approach combines SIS and AIQM to provide a holistic assessment of autonomous transport systems. It considers not only the traditional safety considerations but also the specific implications of AI-powered attacks and the quality attributes of AI systems. This comprehensive assessment allows for a more accurate understanding of the risks and vulnerabilities of ATS and provides a foundation for effective risk mitigation.Transparency and traceability: The proposed SISMECA technique enhances the transparency of assessing the consequences of cyberattacks. By integrating known FMECA/IMECA techniques and developing an ontology model, the approach enables a structured analysis of failure and intrusion modes. This transparency and traceability facilitate a clear understanding of the risks and help identify suitable countermeasures to mitigate those risks.User-centred analysis: By investigating user stories in the maritime, aviation, and space domains, the approach ensures a user-centred analysis of the application of AI in the context of cybersecurity and safety. This user-centric perspective helps identify specific challenges, requirements, and best practices that are relevant to different domains. The insights gained from these user stories contribute to the development of effective assessment techniques and risk mitigation strategies tailored to the needs of different autonomous transport systems.Minimising uncertainty and enhancing risk assessment: The proposed AIQM- and SISMECA-based techniques aim to minimise uncertainty in risk assessment. By decomposing safety and AI quality attributes and applying entropy measures, the approach provides a more accurate and reliable assessment of cybersecurity and safety risks in ATS. This reduction in uncertainty enables stakeholders to make informed decisions and allocate resources effectively to address the identified risks.

The adoption of this approach benefits various stakeholders involved in the development, deployment, and operation of autonomous transport systems:Researchers and academics: The investigation contributes to the theoretical understanding of integrating security, safety, and AI quality in the context of ATS. It provides a methodological base and techniques that can be further explored and expanded upon by researchers and academics in the field.ATS developers and manufacturers: The approach offers a framework and assessment techniques that help ATS developers and manufacturers identify and address cybersecurity and safety risks associated with AI-powered attacks. By integrating these considerations early in the design and development stages, they can enhance the security and reliability of their systems.Regulatory bodies and policymakers: The investigation provides insights and recommendations for regulatory bodies and policymakers in developing standards, guidelines, and regulations for the cybersecurity and safety of autonomous transport systems. It offers a structured approach to assess and evaluate the risks, ensuring that appropriate measures are implemented to protect against AI-driven cyber threats.Operators and service providers: The proposed approach helps operators and service providers of autonomous transport systems to assess and manage cybersecurity risks effectively. By utilising the AIQM and SISMECA-based techniques, they can enhance the protection of their assets, minimise potential disruptions, and ensure the safe and secure operation of their systems.End-users and the public: The adoption of this approach benefits end-users and the general public by increasing the security and safety of autonomous transport systems. By implementing robust cybersecurity measures and considering AI quality attributes, the risk of accidents, disruptions, and unauthorised access to sensitive information can be minimised, thereby ensuring public trust and confidence in these systems.

In summary, the adopted approach of combining SIS- and AIQM-based assessment techniques brings together security, safety, and AI considerations to address emerging challenges, provide holistic assessments, enhance transparency and traceability, minimise uncertainty, and benefit a wide range of stakeholders involved in autonomous transport systems.

While the given methodology has its merits, it is important to consider some counterarguments or potential limitations:Complexity and implementation challenges: The proposed approach involves integrating multiple techniques, methodologies, and models. Implementing such a comprehensive approach may be complex and require significant resources, including expertise, time, and funding. It could pose challenges for organisations with limited capabilities or smaller budgets, potentially limiting the widespread adoption of these techniques.Difficulty in capturing evolving AI threats: The field of AI cybersecurity is rapidly evolving, with new attack vectors and techniques constantly emerging. The proposed approach may face challenges in keeping up with these evolving threats and ensuring that the assessment techniques remain up to date. It may require continuous monitoring and updating to effectively address the dynamic nature of AI-powered attacks.Potential bias and subjectivity in assessments: The integration of AI quality attributes and the subjective nature of assessing risks and criticality could introduce potential bias or subjectivity in the assessment process. Different assessors or organisations may have varying interpretations or weighting of risk factors, which could lead to inconsistent results and decisions.Trade-off between security and usability: Enhancing cybersecurity measures often involves introducing additional layers of security controls, which can impact the usability and user experience of autonomous transport systems. Striking the right balance between security and usability is crucial to ensure that the systems remain efficient and user-friendly, and that they do not hinder their intended functionality.Limited focus on emergent threats and zero-day vulnerabilities: The investigation’s emphasis on known failure and intrusion modes may overlook emergent threats and zero-day vulnerabilities that have not yet been identified or documented. Rapidly evolving cyber threats require continuous monitoring and proactive measures to identify and address new attack vectors effectively.

It is important to acknowledge and address these counterarguments and limitations to refine and improve the proposed approach, ensuring its practicality, scalability, and relevance in real-world applications.

Future research directions are as follows:enhancing models and techniques combining AIQM and SISMECA approaches, including refinement of AI quality and various XMECA templates models to minimise uncertainties and dependencies on expert errors;quantitative assessment of cybersecurity and safety evaluations entropy in the context of attributes decomposition on the application of AIQM and SISMECA;completing tool-based support of AIQM&SISMECA-based techniques for different actors (regulators, developers, operators, and customers) considering their specific responsibility and scenarios of AI-powered protection against AI-powered attacks;developing multi-criteria techniques for choosing countermeasures considering security, reliability, and safety attributes at component and system levels;combining the various assessment methods based on Markov and semi-Markov models to assess embedded and distributed IoT/cloud systems for ATSs [53,54,55].

## Figures and Tables

**Figure 1 entropy-25-01123-f001:**
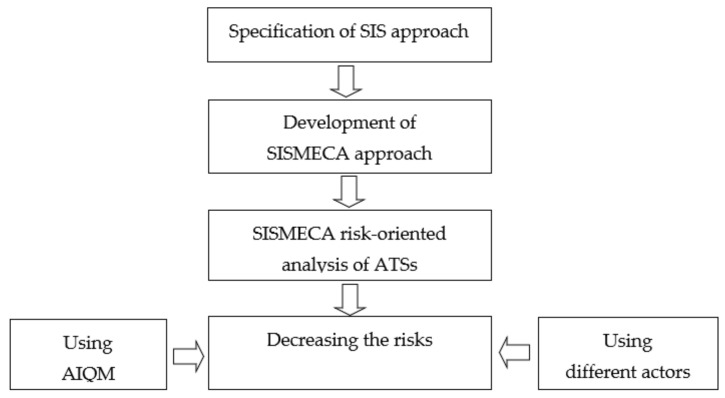
Schematic view of the methodology.

**Figure 2 entropy-25-01123-f002:**
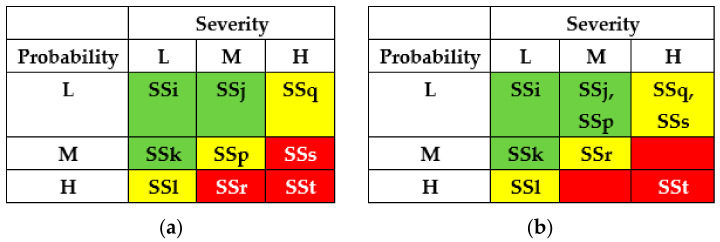
Matrixes of cyber risks. (**a**) Initial matrix. (**b**) Matrix after the implementation of countermeasures.

**Figure 3 entropy-25-01123-f003:**
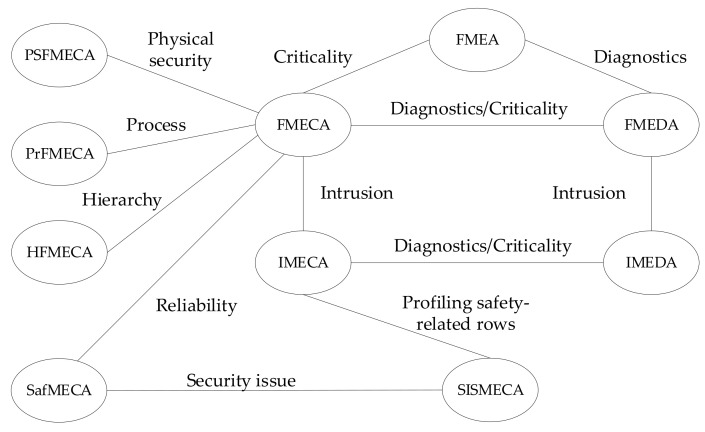
The SISMECA ontology model.

**Figure 4 entropy-25-01123-f004:**
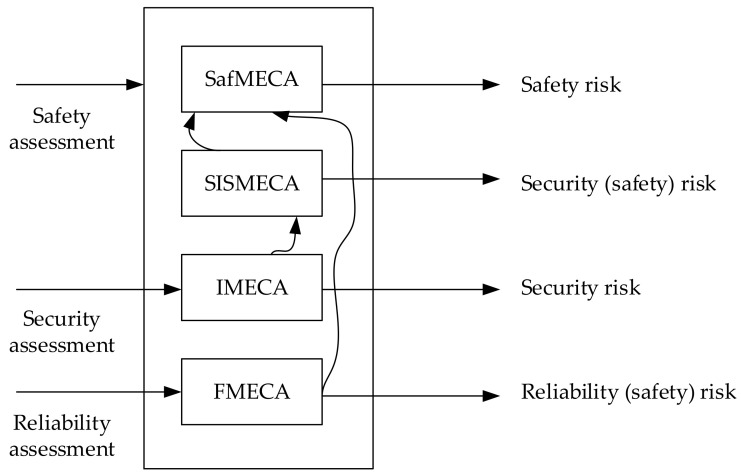
SISMECA as a part of safety assessment techniques.

**Figure 5 entropy-25-01123-f005:**
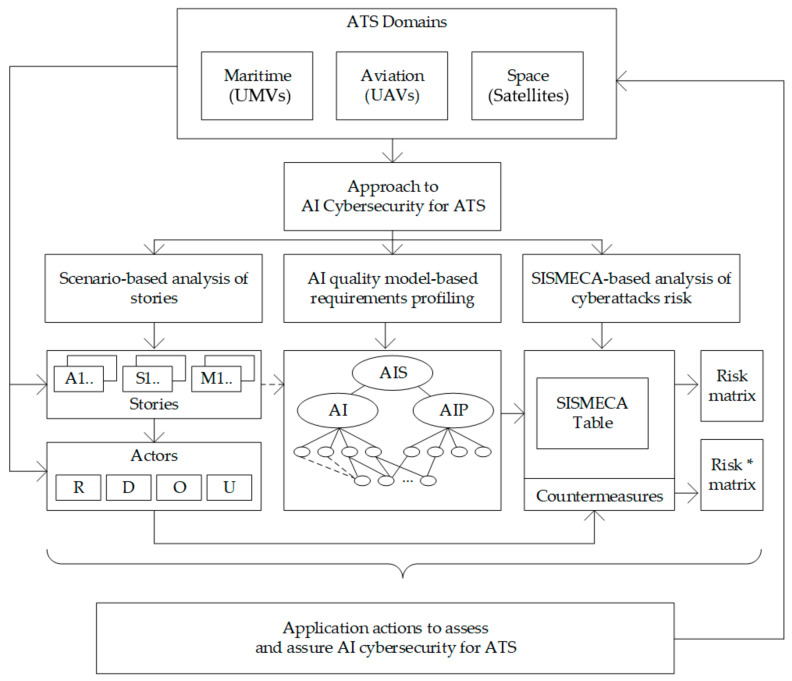
AI quality model and SISMECA-based approach: an overview.

**Figure 6 entropy-25-01123-f006:**
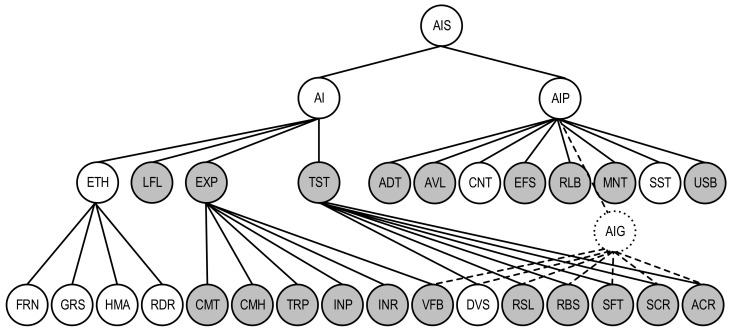
The graph of a general AIS quality model with marked characteristics for the system described in the user story US-ATS.M.

**Figure 7 entropy-25-01123-f007:**
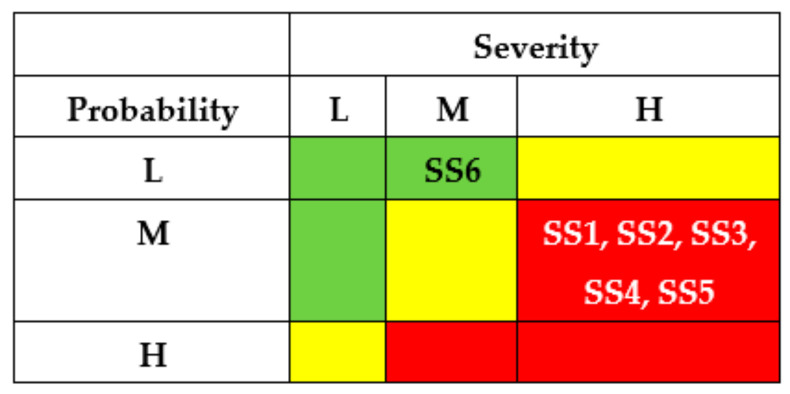
An initial matrix of risks.

**Figure 8 entropy-25-01123-f008:**
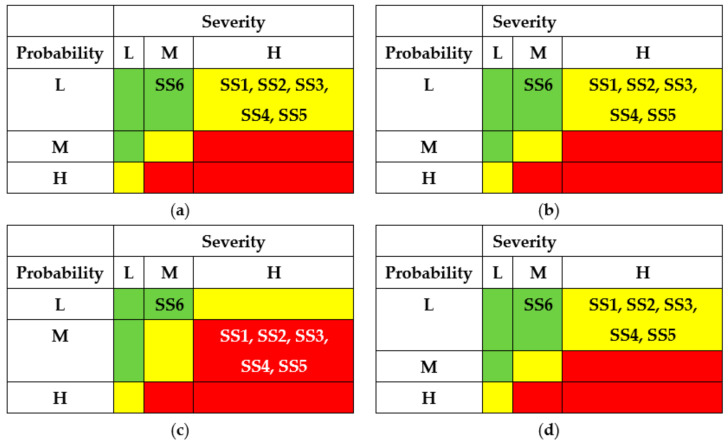
Matrix of risks considering CTMs implemented by DEV (**a**), REG (**b**), OPR, (**c**) and USR (**d**).

**Figure 9 entropy-25-01123-f009:**
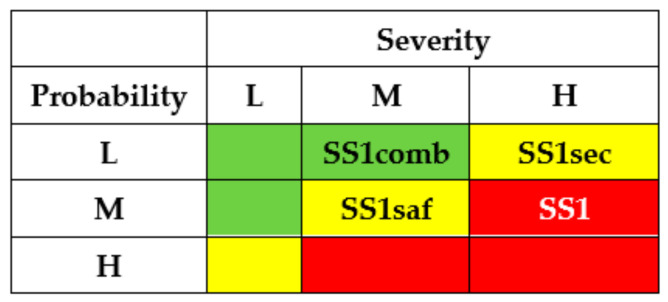
Matrix of risks considering CTMs implemented by DEV.

**Figure 10 entropy-25-01123-f010:**
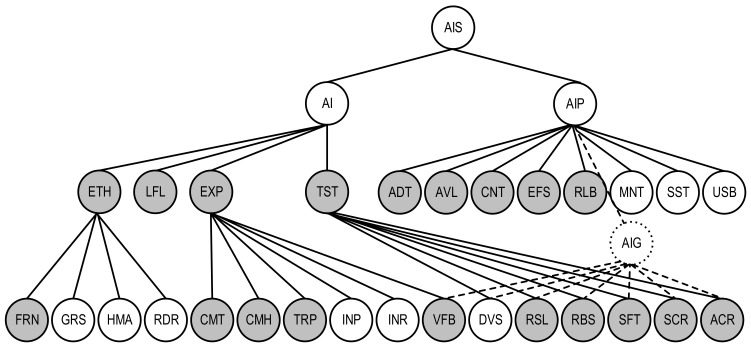
The graph of a general AIS quality model with marked characteristics for the system described in the user story US-ATS.A.

**Figure 11 entropy-25-01123-f011:**
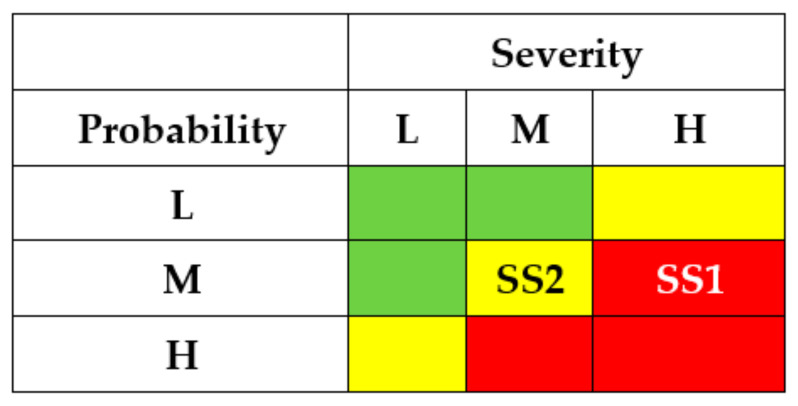
An initial matrix of risks.

**Figure 12 entropy-25-01123-f012:**
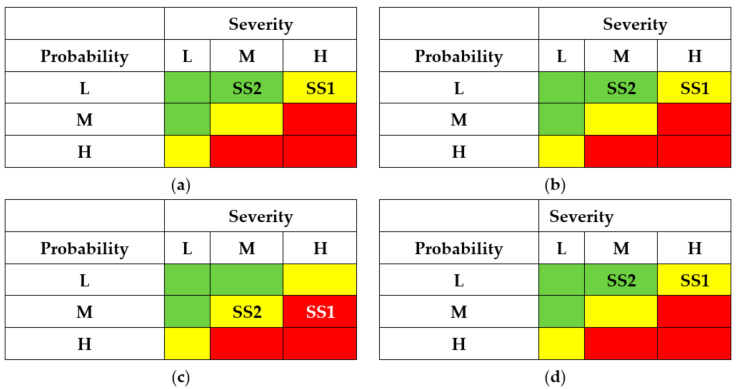
Matrix of risks considering CTMs implemented by DEV (**a**), REG (**b**), OPR (**c**), and USR (**d**).

**Figure 13 entropy-25-01123-f013:**
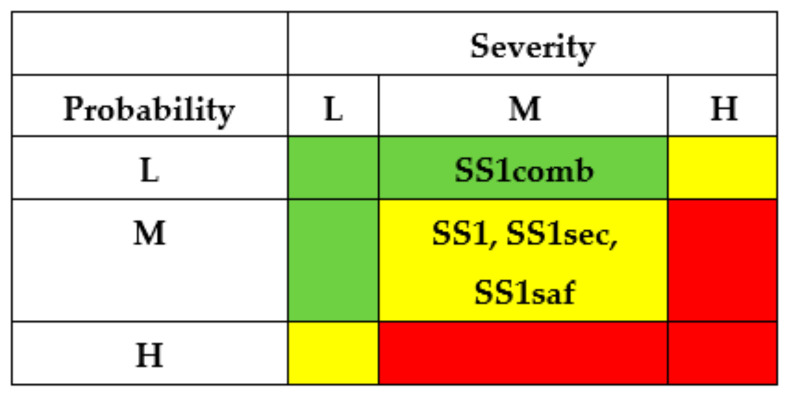
Matrix of risks considering CTMs implemented by DEV.

**Figure 14 entropy-25-01123-f014:**
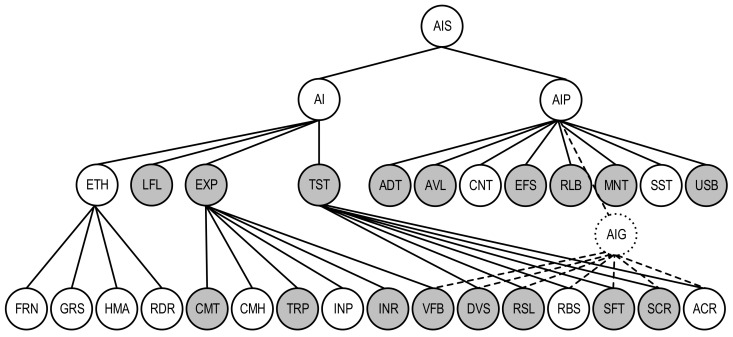
The graph of a general AIS quality model with marked characteristics for the system described in the user story US-ATS.S.

**Figure 15 entropy-25-01123-f015:**
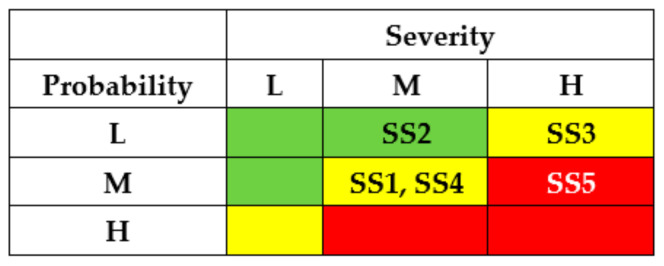
An initial matrix of risks.

**Figure 16 entropy-25-01123-f016:**
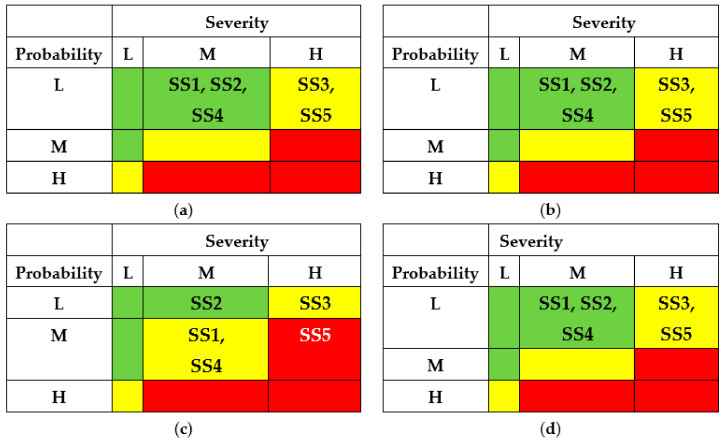
Matrix of risks considering CTMs implemented by DEV (**a**), REG (**b**), OPR (**c**), and USR (**d**).

**Figure 17 entropy-25-01123-f017:**
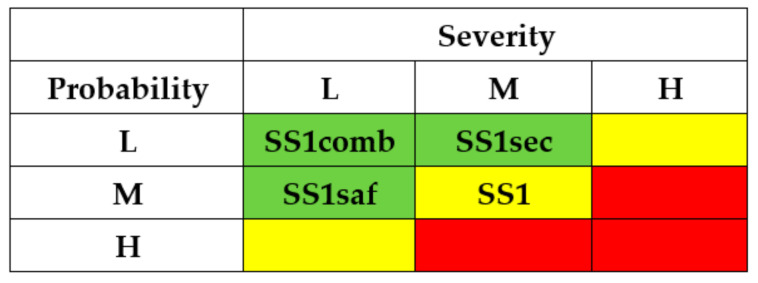
Matrix of risks considering CTMs implemented by DEV.

**Table 1 entropy-25-01123-t001:** AI-powered attacks utilised against various components of the ATS infrastructure.

AI-Powered Attack	Ref.	Technique	Stage	Safety Issues	Targets in the ATS Infrastructure for Attacks					
					On-Board Equipment		GCS		Channel	
					AI-Based	Non- AI-Based	AI-Based	Non- AI-Based	ATS-ATS	ATS-GCS
Intelligent malware	[7]	ChatGPT	Rec	-	-	√	-	√	√	√
PassGAN	[8]	GAN	Acc&Pen	-	-	-	-	√	-	-
Intelligent password brute force attack	[9]	RNN	Acc&Pen	-	-	-	-	√	-	-
Offensive password authentication	[10]	LR, SVM, SVC, RF, KNN, GBRT	Acc&Pen	-	-	-	-	√	-	-
Adversarial malware generation	[11]	GAN	Del	-	√	-	√	-	-	-
Self-learning malware	[12]	K-means clustering	Exp	-	√	√	√	√	-	-
DeepLocker	[13]	DNN	C&C	-	-	-	-	√	-	-

**Table 2 entropy-25-01123-t002:** Potential threats to ATS assets and AI-based intrusion detection/prevention solutions.

Considered Type of ATS	Ref.	Analysed Component of the ATS Infrastructure	Main Contribution	AI-Based Intrusion Detection/ Prevention Solution
UAV	[14]	on-board equipment, UAV–UAV, UAV–GCS	The level of security threats for the various drone categories. A comprehensive taxonomy of the attacks on the Internet of Drones (IoD) network. The results of the review of the recent IoD attack mitigation techniques.	-
	[18]	on-board equipment, UAV–GCS	An autonomous intrusion detection system utilising deep convolutional neural networks (CNNs) to detect the malicious threats invading the UAV efficiently.	√
	[20]	on-board equipment, UAV–GCS	A detection method based on principal component analysis and one-class classifiers for identifying and mitigating spoofing and jamming attacks.	√
	[23]	on-board equipment, UAV–GCS	The results of the experiments to detect, intercept, and hijack a UAV through either de-authentication or jamming.	-
Satellite	[16]	on-board equipment, satellite–satellite, satellite–GCS	The results of the analysis of the past satellite security threats and incidents. The results of segment and sector analysis of satellite security incidents.	-
	[17]	on-board equipment, satellite–satellite, satellite–GCS	Deep Learning (DL)-based flexible satellite network intrusion detection system for detecting unforeseen and unpredictable attacks.	√
	[19]	on-board equipment, satellite–GCS	A lightweight CNN-based detection scheme for detecting barrage, pilot tone, and intermittent jamming attacks against satellite systems.	√
	[21]	GCS, satellite–satellite, satellite–GCS	DL-based network forensic framework for the detection and investigation of cyber-attacks in smart satellite networks.	√
	[22]	GCS, satellite–satellite, satellite–GCS	A robust and generalised DL-based intrusion detection approach for terrestrial and satellite network environments.	√
Maritime Autonomous Surface Ship (MASS)	[15]	on-board equipment, MASS–MASS, MASS–GCS	STRIDE (Spoofing, Tampering, Repudiation, Information Disclosure, Denial of Service and Elevation of Privilege) threat modelling methodology for analysing the accordant risk. A special risk matrix and threat/likelihood criteria for assessing the risk.	-
	[24]	on-board equipment, MASS–MASS, MASS–GCS	The results of the analysis of attack scenarios for MASS cybersecurity risk management. A secure ship network topology for realising MASS operations.	-

**Table 3 entropy-25-01123-t003:** Template of the traditional IMECA table.

Threats (THR)	Vulnerabilities (VLN)	Attacks (ATA)	Effects (EFF)	Probability (PRE)	Severity (SVE)	Criticality (CRT)	Sub-Stories (SSX)
							SS1
							…
							SSN

**Table 4 entropy-25-01123-t004:** Template of an extended IMECA table.

Threats (THR)	Vulnerabilities (VLN)	Attacks (ATA)	Effects (EFF)	Probability (PRE)	Severity (SVE)	Recovery (REC)	Criticality (CRT)	Countermeasures (CTM)	Sub-Stories (SSX)
									SS1
									…
									SSN

**Table 5 entropy-25-01123-t005:** Template of an IMECA table extended with the ACT column.

Threats (THR)	Vulnerabilities (VLN)	Attacks (ATA)	Effects (EFF)	Probability (PRE)	Severity (SVE)	Recovery (REC)	Criticality (CRT)	Countermeasures (CTM)	Actors (ACT)	Sub-Stories (SSX)
										SS1
										…
										SSN

**Table 6 entropy-25-01123-t006:** Template of SISMECA table.

Threats (THR)	Vulnerabilities (VLN)	Attacks (ATA)	Effects on security (EFF_SEC)	Effects on safety (EFF_SAF)	Probability (PRE)	Severity (SVE)	Recovery (REC)	Criticality (CRT)	Countermeasures (CTM), (Actors, ACT)	CRT/CTM	Sub-Stories (SSX)
											SS1
											…
											SSN

**Table 7 entropy-25-01123-t007:** IMECA table for the user story US-ATS.M.

THR	VLN	ATA	EFF	PRE	SVE	REC	CRT (no REC/ REC)	AI-Based CTM (Criticality via PRE Decreasing)				SS
								DEV	REG	OPR	USR	
Activities of hacker centres	Human machine interface weaknesses	Spoofing	Access to the system and critical information	M	H	M	H/M	PRE: L/M; CRT: M/M	PRE: L/M; CRT: M/M	PRE: M/M; CRT: H/H	PRE: L/M; CRT: M/M	SS1
		Tampering	Control and monitor the ship from the shore	M	H	M	H/M	PRE: L/M; CRT: M/M	PRE: L/M; CRT: M/M	PRE: M/M; CRT: H/M	PRE: L/M; CRT: M/M	SS2
		Information disclosure	Damages related to the vessel’s navigation and management	M	H	M	H/M	PRE: L/M; CRT: M/M	PRE: L/M; CRT: M/M	PRE: M/M; CRT: H/H	PRE: L/M; CRT: M/M	SS3
		Denial of service	A ship can be control-less and invisible to the Shore Control Centre	M	H	M	H/M	PRE: L/M; CRT: M/M	PRE: L/M; CRT: M/M	PRE: M/M; CRT: H/M	PRE: L/M; CRT: M/M	SS4
		Elevation of privilege	Access sensitive data about the vessel’s condition, its customers, and passengers	M	H	M	H/M	PRE: L/M; CRT: M/M	PRE: L/M; CRT: M/M	PRE: M/M; CRT: H/M	PRE: L/M; CRT: M/M	SS5
		Repudiation	Distortion of data, which is stored in logs	L	M	M	L/L	PRE: L/L; CRT: L/L	PRE: L/L; CRT: L/L	PRE: L/L; CRT: L/L	PRE: L/L; CRT: L/L	SS6

**Table 8 entropy-25-01123-t008:** Example of the SISMECA table.

Threats (THR)	Vulnerabilities (VLN)	Attacks (ATA)	Effects on Security (EFF_SEC)	Effects on Safety (EFF_SAF)	Probability (PRE)	Severity (SVE)	Recovery (REC)	Criticality (CRT)	Countermeasures (CTM), (Actors, ACT)	CRT/CTM	Sub-Stories (SSX)
Activities of hacker centres	Human Machine Interface weaknesses	Spoofing	Access to the system and critical information	Issue of malicious commands	M	H	L	H/H	(1) CTMsec (2) CTMsaf (3) CTMcomb	M/H M/M L/M	SS1sec SS1saf SS1comb

**Table 9 entropy-25-01123-t009:** IMECA table for the user story US-ATS.A.

THR	VLN	ATA	EFF	PRE	SVE	REC	CRT (no REC/ REC)	AI-Based CTM (Criticality via PRE Decreasing)				SS
								DEV	REG	OPR	USR	
Activities of hacker centres	Navigation system weaknesses	GPS spoofing attacks	Ability to force the UAV to land in a pre-picked zone	M	H	H	H/H	PRE: L/M; CRT: M/M	PRE: L/M; CRT: M/M	PRE: M/M; CRT: H/H	PRE: L/M; CRT: M/M	SS1
		GPS jamming attacks	Loss of control of the UAV	M	M	L	M/M	PRE: L/M; CRT: L/L	PRE: L/M; CRT: L/L	PRE: M/M; CRT: M/M	PRE: L/M; CRT: L/M	SS2

**Table 10 entropy-25-01123-t010:** SISMECA table for the user story US-ATS.A.

Threats (THR)	Vulnerabilities (VLN)	Attacks (ATA)	Effects on Security (EFF_SEC)	Effects on Safety (EFF_SAF)	Probability (PRE)	Severity (SVE)	Recovery (REC)	Criticality (CRT)	Countermeasures (CTM), (Actors, ACT)	CRT/CTM	Sub-Stories (SSX)
Activities of hacker centres	Navigation system weaknesses	GPS jamming attacks	Loss of control of the UAV	Damages caused by uncontrolled UAV	M	M	L	M/M	(1) sec (2) saf (3) comb	M/M M/L L/L	SS1sec SS1saf SS1comb

**Table 11 entropy-25-01123-t011:** IMECA table for the user story US-ATS.S.

THR	VLN	ATA	EFF	PRE	SVE	REC	CRT (no REC/ REC)	AI-Based CTM (Criticality via PRE Decreasing)				SS
								DEV	REG	OPR	USR	
Activities of hacker centres	Communication protocol weaknesses	DDoS attack	Increased network interactions	M	M	L	M/L	PRE: L/M; CRT: L/L	PRE: L/M; CRT: L/L	PRE: MM; CRT: M/L	PRE: L/M; CRT: M/L	SS1
	Power supply devices	DDoS or targeted attack	Power depletion	L	M	H	L/M	PRE: L/L; CRT: L/M	PRE: L/L; CRT: L/L	PRE: L/L; CRT: L/L	PRE: L/L; CRT: L/L	SS2
	Possibility of tampering with the OS/firmware	Tampering attack	A non-functional state known as bricks	L	H	M	M/M	PRE: M/M; CRT: M/M	PRE: M/M; CRT: M/M	PRE: M/M; CRT: M/M	PRE: M/M; CRT: M/M	SS3
	Vulnerabilities of transmitting data between satellites and ground station	Man-In-The-Middle (MITM) attack	Violation of confidentiality and integrity of smart satellites data	M	M	L	M/L	PRE: L/L; CRT: L/L	PRE: L/L; CRT: L/L	PRE: M/L; CRT: M/L	PRE: L/L; CRT: L/L	SS4
		Data manipulation attacks		M	H	M	H/H	PRE: L/L; CRT: M/M	PRE: L/L; CRT: M/M	PRE: M/M; CRT: H/H	PRE: L/L; CRT: M/M	SS5

**Table 12 entropy-25-01123-t012:** SISMECA table for the user story US-ATS.A.

Threats (THR)	Vulnerabilities (VLN)	Attacks (ATA)	Effects on Security (EFF_SEC)	Effects on Safety (EFF_SAF)	Probability (PRE)	Severity (SVE)	Recovery (REC)	Criticality (CRT)	Countermeasures (CTM), (Actors, ACT)	CRT/CTM	Sub-Stories (SSX)
Activities of hacker centres	Power supply devices	DDoS or targeted attack	Power depletion	Uncontrolled drop	L	M	H	M/M	(1) sec (2) saf (3) comb	L/M L/L L/L	SS1sec SS1saf SS1comb

## Data Availability

Not applicable.

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
