# Peer review of "Security-Informed Safety Analysis of Autonomous Transport Systems Considering AI-Powered Cyberattacks and Protection"

_entropy, 2023, doi:10.3390/e25081123_

Round 1

Reviewer 1 Report

Security Informed Safety Analysis of Autonomous Transport Systems Considering AI-Powered Cyberattacks and Protection

The paper presents an overview of modelling for assessing risk against ATS, with a focus on AI powered attacks.   The structure followed is not ideal with there being no explicit literature review. The motivation is provided in Section 1.1, but there is no suitable introduction.   The work is Jargon heavy. While the various acronyms may be familiar to the authors, they need to consider a more general readership, especially given the journal targeted which appears outside the direct scope of the paper.  The authors should consider reworking the text to improve readability for a general audience (e.g. lines 81-91). Key areas in which this can be done, are the reduction of jargon, with the appropriate introduction of acronyms, but limiting them to where it is needed/ Acronyms must be expanded on first use. The second area is in work in on the flow of the document between sections and in letting the reader know what is coming. The used of the multiple Risk plots in the latter half of the paper does break it up and reduce the readability.  Risk matrixes, break up text making follow hard to follow, consider reducing/using subfigures. these could be scaled down in size without loss of readability.

Overall, the language while stilted in places, is semantically and grammatically sound, with only a few minor typos identified. A stylistic issue to consider is the overly frequent use of bulleted lists, (sometimes nested within a numbered list). Some of these could be moved to tables or reworked as narrative text. Continuing with a stylistic feedback, the authors should avoid jumping from a main to a subsection e.g. 3.1 -> 3.1.1 . In the case stated, is this subsubsection needed? This is done in several places in the paper. Generally, style used is that a section will have some introductory material before a subsection starts (or subsubsection as appropriate)

The paper does not come through with a clear focus after wading though the three cases for air marine and space.  The conclusion could be improved in this regard. It is also difficult for a reader to necessarily follow the logic when there are no ‘specific’ exemplar values provided, having to rely on the jumps provided by the authors. One possibility of simplifying things is to reduce the focus of the work. For example, the focus could be on the development of the SIS, or on the AI elements. As it stands there appears to be an overflow of information being provided and the coverage being quite broad. This in turn means there is not sufficient space for examples.

The paper is an interesting and valid topic, but the authors need to polish this work prior to publication. Part of this should be consideration of if Entropy really is the best venue for publishing the work.

Further specific comments follow:

line 39 - expansion of ATS. This is expanded in the abstract but should be done so in the main body of the paper as well. This applies to the use of other acronyms in the paper. Define /expand on first use.

line 68 - consider "were subdivided in the following ways" or "were subdivided into the following:"

line 72 - consider "of ATS infrastructure". "the" implies specifics and you have not introduced a specific infrastructure to the discussion yet.

Table 1 could stage be written out to help table stand on its own? Why is safety issues included when there are no cases? Avoid hyphenating terms such as "On-board equip-ment" could this not be presented as "On-board\nequipment" consider carefully the layout of the table regarding column spacing and wrapping of text. Ref as a second column, as it is the techniques that are important, alternatively have stage as the primary column since it appears ordered by that?

Table 2

would it not be best to order this table by the type of ATS, dealing with teach category at a time. This should be the primary column too.

Narrative around tables 1 and 2 - while the bullet points summarise the key points these may be more readable as text that is written in more of a narrative manner.

lines 241-246 - would these not be better as a numbered list. This is somewhat repeated in Figure 1

line 299 compare "Security informed Safety (SiS)" with " Cybersecurity informed safety (CSiS)". Consistency is important.

The value of table 3 is unclear. Consider the abbreviations in parentheses () rather than as a comma, . These acronyms could be better defined in text and then used consistently  in what follows, e.g. lines 371-378

Given the presence of Table 5 as an extended (and thus improved IMECA table), what is the value of Table 4? 

The work could possibly be condensed by presenting the final model /table rather than the specifics of each evolutionary step. The accompanying discussion can still provide the link and argument without breaking up the flow.

Figure 2, would the labels for pre/post countermeasures not be better suited in the subfigures?

Figure 5 - typo (Satellits), is "ship robots" the best term I have come across is UMV in literature.

Where is "US-ATS.M" defined? (This does make more sense with subsequent reading, but the reader should not be left to deduce terms such as this.)

section 3.1.2 could the bullet points not possibly be better presented as a table?

Fig 6 - the dashed lines should be explicitly explained.

While Figure 7 presents the initial risk matrix, it is difficult to follow how the following 4 figures are derived given the lack of any matrices with values. This is left to the reader to trust the author's narrative.

The same applies in the 3.2 Aviation and 3.3 Space domains. It’s worth noting that these are dealt with out of order to which the domains are presented in Figure 5

line 725 - inconsistent ordering of these domains again.

Abbreviations:

Confirm to Journal style relating to the presentation of this. Consider not using centering, but rather traditional left alignment.

Consistency is needed in use of capitalisation eg. ATA vs. ATS ; CRT vs DNN

Is ATA correct vs ASA?

Com&Con - common terminology is C2 or C&C

References:

Check article titles for consistency in capitalisation.

Consistency in the use of abbreviations of journal titles.

Consistency of citation eg [1,2,3] vs [27],28]

4 - be consistent regarding journal name abbreviations

5 - Ai -> AI

6 - disrup tor -> one word?  Is this really the best reference for the point?

15 - Journal abbreviated title

24 Proceedings of CEUR-WS. Title in full ?

43-50 - Author checks. E.g. Should this be EU Parliament, and then the title.

Generally good, but readability could be improved, see comments above. Very fewtypos/grammar issues

Author Response

Thank you for your feedback on our paper which has greatly helped us to improve the quality and presentation of the paper. We have carefully addressed all the comments raised by the reviewers in this revised version. We explain below how the comments of each reviewer have been carefully addressed. The manuscript has been improved in terms of the English language and the changes introduced while addressing the comments of Anonymous Reviewers. All of them are marked in colour.

We sincerely hope you and the Reviewers will be satisfied with all our revisions.

Thank you for your consideration.

Sincerely,

Authors

Reviewer 2 Report

This is a very interesting paper that has both width and depth. It appears well researched and logical. I only have a few points.

The paper is very long and I wonder if it can be turned into 2 papers. One paper would set the scene and be more focused on the type of cyber related problems and how they are to be overcome, with attention being given to developing some of the underpinning arguments and linking more widely with business continuity planning for example.

Some of the threats identified can be expanded and also, the bullet points cited could where possible be expanded also. It is not a good idea to have too many bullet points in a paper.

One of the issues appears to be related to academic theory, which is rather thin. How are the points cited related to and explained in a more theory driven line of argument?

It is not that clear why this approach has been adopted and who will benefit from it. This can all be explained in greater depth.

The outcome can be expanded in terms of the application of the framework and this may be relevant as regards the second paper.

If it is decided to keep the paper as it is, possibly just reflect on the points above and see if some aspects can be explained better. It is always good to justify points and provide counterarguments where necessary.

Possibly the discussion can be expanded slightly and some of the pitfalls can be highlighted as well as the management implications, which also need more attention.

Author Response

(The authors gave the same response as above.)

Reviewer 3 Report

1- the paper briefly touches upon the potential use of AI-based cybersecurity tools in cyberattacks on ATS. Including real-world examples or case studies where AI-based tools have been exploited or misused in attacks on ATS would strengthen the discussion. This would provide readers with concrete instances that highlight the risks and implications associated with these tools.

2- I believe that the inclusion of some related papers could further enhance the depth and breadth of your research. I would like to suggest the following papers for your consideration:

1- Ziaul Hasan, Hassan r. Mohammad, & Maka Jishkariani. (2022). Machine Learning and Data Mining Methods for Cyber Security: A Survey . Mesopotamian Journal of CyberSecurity, 2022, 47–56. https://doi.org/10.58496/MJCS/2022/006

2- Yamin, Muhammad Mudassar, et al. "Weaponized AI for cyber attacks." Journal of Information Security and Applications 57 (2021): 102722.

3- Taseer Muhammad, & Hamayoon Ghafory. (2022). SQL Injection Attack Detection Using Machine Learning Algorithm. Mesopotamian Journal of CyberSecurity, 2022, 5–17. https://doi.org/10.58496/MJCS/2022/002

Author Response

(The authors gave the same response as above.)

Round 2

Reviewer 1 Report

Overview

This paper has been revised but is still not an easy ready.   The authors are encouraged to consider what could be restructured/trimmed (such as the empty tables showing the evolution?) to allow for a core focus.  The work presents an interesting idea but its difficult to consider a critical evaluation and comparison of this work and the impact there of.

Keywords:

Authors should carefully consider if these are all appropriate and/or required.

Gratuitous citations on lines 68,71 & 73 - are all these references actually required.  The majority are not used anywhere else.   The intention it to provide examples to backup the claims, but as on line 71 , twenty-one citations is likely excessive, especially considered there is a lack of discussion of specifics. In the case od line 71 a number of these papers are self-citations to authors 1-4. While there is nothing wrong with referring to prior work, it must be directly relevant to the point being made.

line 70 - "AI and AI platform (AIP" consider "AI, and AI platforms (AIP"

line 74 - Kaloudi et al. [5] reads in a more natural manner.

Table 2 should not be split over pages, some minor editing  could reduce table size substantially ( col 1 rotation of labels), reduce columns 2,5 and widen 3/4

lines 238-248 - is there a need for principle to be abbreviated? or even used rather than a numbered list

lines 346,349 - these sentences could be joined as a paragraph.

line 552 - US is used for usability, but this is not conforming with other uses which is USB.

Table 8 - avoid hyphenation of Machine; table should not be split over pages

Figure 7 - why is the dimensions/scale of this different to similar figures? (Similar point applies to Fig 9,11,13,15,17,)

Figure 8 - consider the detail naming in the subfigure as used elsewhere

line 700 -  USB (usability). Acronyms must be expanded/introduced in text. The reader cannot be expected to have to refer to the list for looking them up( although it does provide a useful reference)

Figures:

Authors should carefully check the size and quality of images, some could be significantly reduced in size and still maintain readability.

Acronyms

USB - Usability - this is a case where the overloading of a commonly used acronym with a different meaning is unnecessary jargon. That said this is used primarily in the Figures for conciseness. It is first used in fig 6, and there is no explanation for expansion in text, with the first appearance in text on line 700

References

[44] The authors of this should be checked. likely institutional author of Council of Europe

[45/46] Check authors.

Grammar is good, but the writing while correct is a difficult read. This is due to a fairly jargon heavy work littered with (over) use of acronyms.

Author Response

Dear Anonymous Reviewer,

Thank you for your second feedback on our paper which has even greatly helped us to improve the quality and presentation of the paper. We have carefully addressed all the comments raised by the reviewers in this second revised version. We explain below how the comments of each reviewer have been carefully addressed. The manuscript has been improved in terms of the English language as well and the changes introduced while addressing the comments of other Anonymous Reviewers. All of them are marked in colour.

We sincerely hope you and the Reviewers will be satisfied with all our revisions.

Thank you for your consideration.

Reviewer 2 Report

Good overall improvement.

Author Response

Dear Anonymous Reviewer,

Thank you for your valuable comment. We tried to do our best. Please, see the final improvements.

The manuscript has been improved in terms of the English language as well and the changes introduced while addressing the comments of other Anonymous Reviewers. All of them are marked in colour.